# End-to-End Autoregressive Image Generation with 1D Semantic Tokenizer

**Wenda Chu**[1,2]  **Bingliang Zhang**[1,2]  **Jiaqi Han**[1,3]  **Yizhuo Li**[1]  **Linjie Yang**[1]  **Yisong Yue**[1]  **Qiushan Guo**[1]

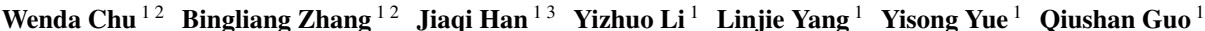

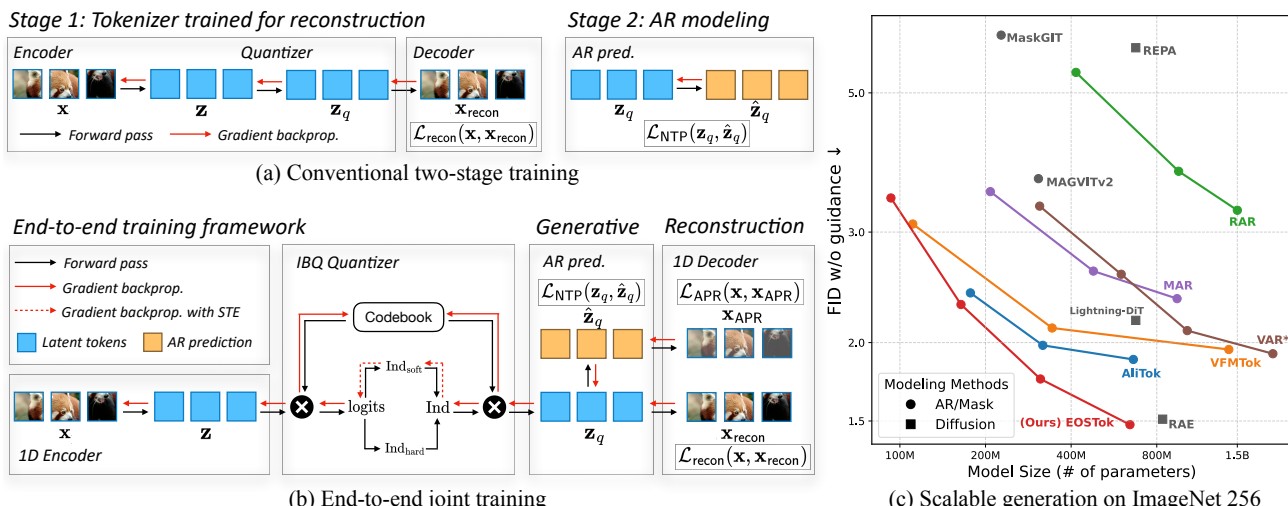

*Figure 1.* **Overview of EOSTok that jointly trains an 1D vision tokenizer and an autoregressive generative model.** (a) Conventional paradigms employ a two-stage training strategy. The tokenizer is trained on a reconstruction task. An autoregressive generative model is then trained on the token sequences encoded by the frozen tokenizer. (b) Our training framework jointly optimizes the tokenizer and the AR model, enabling end-to-end generative supervision to guide the tokenizer. (c) Our EOSTok model enjoys a superior scaling ability on the ImageNet 1K 256×256 generation benchmark. We compare EOSTok to generative models with AR, mask, or diffusion modeling. Numbers are collected without guidance except for those with (*).

## Abstract

Autoregressive image modeling relies on visual tokenizers to compress images into compact latent representations. We design an end-to-end training pipeline that jointly optimizes reconstruction and generation, enabling direct supervision from generation results to the tokenizer. This contrasts with prior two-stage approaches that train tokenizers and generative models separately. We further investigate leveraging vision foundation models to improve 1D tokenizers for autoregressive modeling. Our autoregressive generative model achieves strong empirical results, including a state-of-the-art FID score of 1.48 without guidance on ImageNet 256×256 generation.

## 1. Introduction

Inspired by the remarkable success of autoregressive (AR) modeling in large language models (Brown et al., 2020), recent work has increasingly explored autoregressive visual generation via discrete visual tokens, positioning AR models as a compelling alternative to diffusion-based approaches (Van Den Oord et al., 2017; Sun et al., 2024). However, most existing image generative models rely on 2D grid-structured tokenizers that preserve the spatial layout of pixel patches. Such 2D tokenizations induce inherently bidirectional dependencies among tokens, which are fundamentally misaligned with the unidirectional factorization required by raster-order autoregressive modeling.

Two broad classes of approaches have been explored to address this issue. One line of work retains 2D aligned tokenizers but designs new AR modeling schemes beyond raster-scan ordering, such as masked AR generation with random ordering (Li et al., 2024; Chang et al., 2022) or next-scale prediction with multi-scale 2D tokenization (Tian et al., 2024). In contrast, another line of work (Yu et al., 2025b; Ge et al., 2023; Bachmann et al., 2025; Wen et al., 2025) seeks to build 1D tokenizers suitable for generative modeling. For example, TiTok (Yu et al., 2025b) concatenates image

[1]ByteDance Seed  [2]California Institute of Technology  [3]Stanford University.  Correspondence to: Wenda Chu <wchu@caltech.edu>.

*Proceedings of the 43rd International Conference on Machine Learning*, Seoul, South Korea. PMLR 306, 2026. Copyright 2026 by the author(s).

patches with learnable query tokens, forcing these query tokens to compress global visual information into a compact 1D latent representation without imposing an explicit 2D spatial prior. This architecture was originally designed (Yu et al., 2025b) for the purpose of high compression rates (e.g., 32 tokens), which trades off reconstruction quality for the ease of generative modeling. However, we argue that the removal of 2D structural dependency paves the way for designing visual tokens that naturally support vanilla autoregressive modeling, which does not necessarily rely on aggressive compressions.

To achieve this goal, we introduce an ***end-to-end single-stage training paradigm*** that jointly optimizes for reconstruction and autoregressive generative modeling. Unlike the conventional training paradigm that trains tokenizers only for reconstruction, we propose to jointly optimize reconstruction and generation, which enables direct generative feedback to the tokenizer. Moreover, we find that the next-token-prediction loss on *discrete tokens* cannot determine the final generative quality in the *pixel space*. To bridge this gap, we design an ***Autoregressive Prediction Reconstruction*** (APR) loss that decodes the teacher-forcing prediction of the AR model into pixel space and computes a reconstruction loss. We show our end-to-end training pipeline improves the final generation quality and makes the latent space more autoregressive-predictable.

Furthermore, inspired by previous works (Yao & Wang, 2025; Yu et al., 2025c) that use vision foundation models (VFM) to regulate the latent space of 2D tokenizers and improve diffusion models, we investigate how to effectively inject semantic VFM representations into a 1D tokenizer. However, directly aligning the 1D sequential latent space to 2D VFM representations (Yao & Wang, 2025) forces it to degenerate to a raster-ordered, patch-aligned sequence, leading to suboptimal performance. We thus propose an ***implicit alignment*** strategy that aligns hidden patch embeddings to VFM representations instead. This strategy distills global semantic information from VFMs into the sequential latent space without enforcing spatial structure, resulting in substantially improved generative quality.

Putting it together, we present EOSTok, an **E**nd-to-end **O**ne-dimensional **S**emantic **Tok**enizer that jointly optimizes reconstruction, generation, and semantic alignment. Our method is effective and easily scalable, as the EOSTok-H model with 644M parameters achieves a state-of-the-art FID score of 1.48 without guidance on ImageNet-1K 256×256 generation.

## 2. Related Work

**Image tokenizers.** Image tokenizers compress high-dimensional images into compact low-dimensional latent

representations. Variational autoencoders (Kingma et al., 2013; Rombach et al., 2022) (VAEs) typically consist of an encoder that maps images $\mathbf{x}$ to latent embeddings $\mathbf{z} = \mathcal{E}(\mathbf{x})$ and a decoder that reconstructs the image as $\hat{\mathbf{x}} = \mathcal{D}(\mathbf{z})$, which are optimized using reconstruction and variational KL loss. While VAEs learn a continuous latent space, VQ-VAEs (Van Den Oord et al., 2017; Razavi et al., 2019; Esser et al., 2021) map images to discrete tokens by adding a vector quantization module after the encoder, defining latent tokens as $\mathbf{z} = \mathcal{Q}(\mathcal{E}(\mathbf{x}))$. Later works improve VQ-VAE by applying residual quantization (Lee et al., 2022) and dynamic quantization (Huang et al., 2023), and decreasing code dimension (Sun et al., 2024; Yu et al., 2021). To scale the codebook size, recent approaches (Yu et al., 2023; Mentzer et al., 2023; Zhu et al., 2024; Shi et al., 2025a) design new quantization methods to increase codebook utilization. Recently, 1D tokenizers (Yu et al., 2025b; Ge et al., 2023; Bachmann et al., 2025; Wen et al., 2025) that encode 2D images into 1D sequences have gained more attention, which are further explained in the following paragraph.

**Autoregressive visual generation.** Early works on autoregressive modeling of images (Lee et al., 2022; Sun et al., 2024) predict image tokens in a raster-scan order, using tokenizers with latent spaces spatially aligned to 2D images. However, this strategy creates bidirectional dependencies among tokens, making it misaligned and thus suboptimal for causal autoregressive generation. To address this issue, MaskGIT (Chang et al., 2022) and MAR (Li et al., 2024) propose masked AR modeling with bidirectional attention, while VAR (Tian et al., 2024) proposes autoregressive modeling for next-scale prediction. Recently, TiTok (Yu et al., 2025b) and SEED (Ge et al., 2023) propose to extract 1D representations of images using learnable query tokens. The vision transformer (Dosovitskiy et al., 2021) (ViT) encoder appends these query tokens $\mathbf{q}$ to the 2D image patches $\mathbf{x}_{\text{patch}}$ and outputs only the query tokens $\mathbf{z} = \mathcal{E}_\phi(\mathbf{x}_{\text{patch}} \oplus \mathbf{q})$. To reconstruct images, the ViT decoder concatenates learnable mask tokens $\mathbf{m}_{\text{patch}}$ with the latent tokens, transforming the mask tokens back to images $\hat{\mathbf{x}} = \mathcal{D}_\psi(\mathbf{m}_{\text{patch}} \oplus \mathcal{Q}(\mathbf{z}))$. ALIT (Duggal et al., 2025), FlexTok (Bachmann et al., 2025), KARL (Duggal et al., 2026), and Semanticist (Wen et al., 2025) further introduce variable-length 1D tokenizers, enforcing important information to be represented by earlier tokens.

**Representation from vision foundation models.** Leveraging semantic features from pre-trained vision foundation models (VFMs), e.g., DINO (Oquab et al., 2023; Siméoni et al., 2025) and CLIP (Radford et al., 2021), have achieved remarkable success in improving diffusion models. A line of work (Yu et al., 2025c; Yao & Wang, 2025; Leng et al., 2025) aligns the latent space (VA-VAE (Yao & Wang, 2025)) or intermediate layers of diffusion models (REPA (Yu et al., 2025c)) to the corresponding VFM representations. Specif-

ically, let $f$ be the pre-trained VFM and $\mathbf{y} = f(\mathbf{x})$ be the VFM representation. The REPA loss is defined as $\mathcal{L}_{\mathsf{REPA}} = -\frac{1}{N} \sum_{n=1}^{N} \mathsf{sim}(h_{\boldsymbol{\omega}}(\mathbf{h}^{[n]}), \mathbf{y}^{[n]}))$, where $h_{\boldsymbol{\omega}}$ is a learnable MLP projector, $n$ is the patch index, and $\mathbf{h}$ is either the latent vector of the VAE (Yao & Wang, 2025), or the hidden state from an early layer of diffusion models (Yu et al., 2025c). In contrast, another line of work (Zheng et al., 2025b; Shi et al., 2025b; Bi et al., 2025; Chen et al., 2025) directly substitutes the VAE encoder with a frozen, pre-trained vision encoder, optionally adding a lightweight learnable adaptation module.

## 3. Method

Our goal is to produce a 1D vision tokenizer that compresses images to a sequential latent representation $\mathbf{z}$ that facilitates autoregressive modeling, which predicts $p(\mathbf{z}_n \mid \mathbf{z}_{<n})$. In this section, we first introduce our 1D ViT tokenizer architecture, and then introduce how end-to-end joint training (Section 3.2) and semantic VFM representation (Section 3.3) help us achieve this goal.

### 3.1. 1D Vision Transformer Tokenizer

Our tokenizer is a ViT-based autoencoder with a discrete, 1D sequential latent space, which is designed following a similar structure as TiTok (Yu et al., 2025b). As shown in Figure 2, 2D-grid image patches $\mathbf{x}_{\mathsf{patch}} \in \mathbb{R}^{N \times D}$ are flattened and concatenated with $L$ learnable query tokens $\mathbf{q} \in \mathbb{R}^{L \times D}$. This sequence is passed to a causal ViT encoder $\mathcal{E}_{\phi}$, yielding $[\mathbf{h}_{\mathsf{Enc}}, \mathbf{z}] = \mathcal{E}_{\phi}([\mathbf{x}_{\mathsf{patch}}, \mathbf{q}])$, where the hidden patch embedding $\mathbf{h}_{\mathsf{Enc}}$ is discarded and only the 1D latent representation $\mathbf{z}$ is retained. We employ vector quantization with IBQ (Shi et al., 2025a) and quantize the latent code to $\mathbf{z}_q = \mathcal{Q}(\mathbf{z})$. Symmetrically, the 1D decoder $\mathcal{D}_{\psi}$ is designed to take in $[\mathbf{z}_q, \mathbf{m}_{\mathsf{patch}}]$ and reconstruct $[\varnothing, \mathbf{x}_{\mathsf{recon}}]$, where $\mathbf{m}_{\mathsf{patch}} \in \mathbb{R}^{N \times D}$ are identical mask tokens.

**Quantizer.** The IBQ (Shi et al., 2025a) quantizer computes $\mathsf{logits} = [\mathbf{z}^T \mathcal{C}_1, \ldots, \mathbf{z}^T \mathcal{C}_K] \in \mathbb{R}^K$ with $\mathcal{C} \in \mathbb{R}^{K \times D}$ being the codebook. To enable gradient propagation, IBQ implements a straight-through estimation trick to compute code indices, i.e.,

$$\mathsf{Ind} = \mathsf{onehot}(\arg\max \mathbf{p}) + [\mathbf{p} - \mathsf{stopgrad}(\mathbf{p})], \quad (1)$$

where $\mathbf{p} = \mathsf{softmax}(\mathsf{logits})$. $\mathbf{z}_q = \mathsf{Ind}^T \mathcal{C}$ is the quantized output representation.

This 1D ViT architecture explicitly eliminates the 2D spatial prior from the tokenization process. Such decoupling allows us to design an image tokenizer that is inherently compatible with autoregressive generation. To this end, we propose an end-to-end training framework (Section 3.2) that jointly optimizes reconstruction and autoregressive generation. Furthermore, we investigate how incorporating semantic features from vision foundation models into 1D

*Table 1.* **Joint training with APR loss prevents latent space collapsing and improves AR generation quality.** Numbers are reported by training a EOSTok-L model for 50 epochs. Images are generated *without* classifier guidance and the prediction accuracy of the AR model is evaluated on the validation dataset. Code usage counts the code with a frequency of more than $5\%/K$ on the validation dataset.

|  | rFID ↓ | gFID ↓ | AR Acc. | Code Usage |
|---|---|---|---|---|
| Baseline | 1.09 | 3.82 | 11.8% | 99.8% |
| Vanilla E2E | 4.92 | 8.01 | 30.2% | 51.8% |
| + APR loss | **1.02** | **3.32** | 11.9% | 99.7% |

tokenizers can improve both reconstruction quality and generative performance (Section 3.3).

### 3.2. Joint Training of Reconstruction and Generation

Consider the task of training an image tokenizer with encoder $\mathcal{E}_{\phi}$ and decoder $\mathcal{D}_{\psi}$. The tokenizer is optimized by

$$\mathcal{L}_{\mathsf{VQVAE}}(\boldsymbol{\phi}, \boldsymbol{\psi}) = \mathcal{L}_{\mathsf{recon}}(\mathbf{x}, \mathcal{D}_{\psi}(\mathbf{z}_q)) + \lambda_{\mathsf{reg}} \mathcal{L}_{\mathsf{reg}}, \quad (2)$$

where $\mathbf{z}_q = \mathcal{Q}(\mathbf{z}) = \mathcal{Q}(\mathcal{E}_{\phi}(\mathbf{x}))$ is the quantized latent representation of the image $\mathbf{x}$. $\mathcal{L}_{\mathsf{recon}}$ is a combination of $L_1/L_2$ loss, perceptual loss (Zhang et al., 2018), and GAN loss (Esser et al., 2021); while $\mathcal{L}_{\mathsf{reg}}$ regularizes the quantizer $\mathcal{Q}$, including commitment loss (Zhang et al., 2018), entropy loss (Yu et al., 2023), and etc.

We study enhancing visual tokenizers for AR generation by jointly optimizing it for reconstruction and generation and enabling end-to-end generative supervision directly from AR modeling. The conventional paradigm trains a tokenizer for reconstruction in the first phase and a generative model in the second phase, with the tokenizer parameters frozen. However, we argue that freezing the tokenizer in the second phase prevents it from learning a representation better suited to the generative task. This motivates us to design a ***single-stage end-to-end*** training pipeline that jointly optimizes the tokenizer and the generative model from scratch. The overall loss function can be written as

$$\mathcal{L}_{\mathsf{E2E}}(\boldsymbol{\phi}, \boldsymbol{\psi}, \boldsymbol{\theta}) = \mathcal{L}_{\mathsf{VQVAE}}(\boldsymbol{\phi}, \boldsymbol{\psi}) + \lambda_{\mathsf{NTP}} \mathcal{L}_{\mathsf{NTP}}(\boldsymbol{\phi}, \boldsymbol{\theta}), \quad (3)$$

where $\boldsymbol{\phi}, \boldsymbol{\psi}, \boldsymbol{\theta}$ are parameters of the VAE encoder $\mathcal{E}_{\phi}$, the decoder $\mathcal{D}_{\psi}$, and the AR generative model $\mathcal{G}_{\theta}$, respectively. $\mathcal{L}_{\mathsf{NTP}}$ is the next token prediction loss, which depends on not only the AR model $\boldsymbol{\theta}$ but the encoder $\boldsymbol{\phi}$ as well.

**Gradient propagation.** During joint training, the AR model is trained on quantized tokens that are not naturally differentiable with respect to the tokenizer. To enable gradient supervision from AR modeling, we modify the embedding layer of our AR model so that it takes in probability $\mathsf{Ind} \in \mathbb{R}^{L \times K}$ and compute the embedding as $\mathbf{h} = \mathsf{Ind}^T \mathsf{Embed}$ instead of a look-up operation. This enables gradient backward of $\mathcal{L}_{\mathsf{NTP}}$ to the VAE encoder and the codebook.

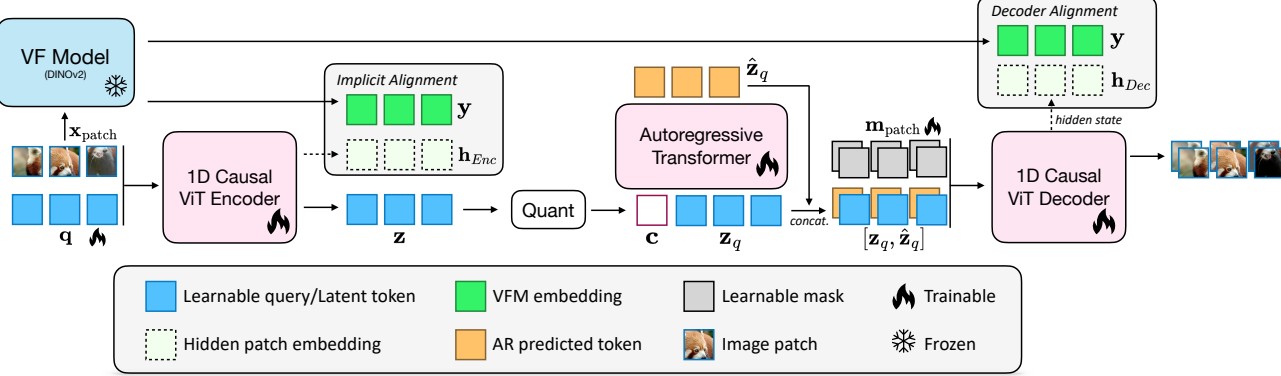

*Figure 2.* **Overall training pipeline of EOSTok.** The 1D ViT tokenizer and the autoregressive generative model are jointly trained. The training objectives include (1) *reconstruction loss* $\mathcal{L}_{\text{VQ-VAE}}$ that optimizes the tokenizer; (2) *generative loss* (next token prediction) $\mathcal{L}_{\text{NTP}}$ that optimizes AR model and supervises the latent space; (3) *AR prediction reconstruction* (APR) loss (Section 3.2) $\mathcal{L}_{\text{APR}}$ that decodes next token predictions to the pixel space and provides end-to-end generative feedback; and (4) *representation alignment loss* $\mathcal{L}_{\text{align}}$ (Section 3.3) that aligns both the latent space and decoder to semantic VFM embeddings.

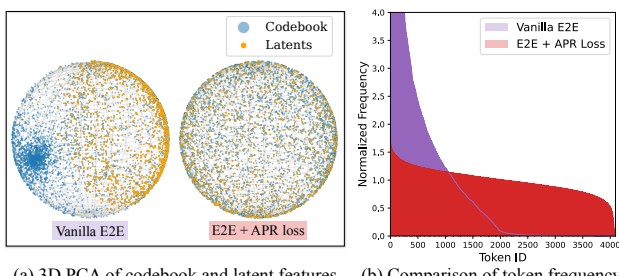

(a) 3D PCA of codebook and latent features    (b) Comparison of token frequency

*Figure 3.* **Next-token-prediction loss hacking the latent space.** (a) NTP loss encourages the tokenizer to use only a subset of tokens in the codebook, and causes the codebook to collapse. This is alleviated by the APR loss, which regulates the latent space by end-to-end feedback from the pixel space. (b) The tokenizer hacks the NTP loss by using very few tokens.

**Gap between NTP loss and generation quality.** The biggest challenge of jointly training tokenizer and AR model is that the next-token-prediction loss is not an end-to-end objective. The NTP objective is defined on discrete token space that is constantly changing during training, and it cannot reflect the final generation quality in the pixel space.

To demonstrate this gap, we train a model using the joint training framework, which we refer to as *Vanilla E2E*. As shown in Table 1, supervising the tokenizer with $\mathcal{L}_{\text{NTP}}$ significantly boost the AR prediction accuracy (from 10.8% to 30.2%). However, the NTP loss hacks the tokenizer by *collapsing the latent space into using very few tokens*, resulting in a dramatic decrease in codebook usage, rFID, and gFID.

We further visualize its codebook by a principal component analysis and projecting it onto a 3D sphere in Figure 3. As shown in Figure 3a, the codebook distribute unevenly in the latent space, and the latent embedding of images could only match a small fraction of codes. This is corroborated by Figure 3b, where the frequency distribution is highly skewed towards a small subset of tokens.

**Bridging NTP and generative quality.** To solve this issue, we propose a simple yet effective *autoregressive prediction reconstruction (APR)* loss to bridge the gap between NTP loss and the overall generation quality. We take the predicted tokens of AR model in teacher forcing, decode them directly to pixels using the decoder, and match with the ground-truth image. The loss function can be written as

$$\mathcal{L}_{\text{APR}}(\phi, \psi, \theta) = \|\mathbf{x} - \mathcal{D}_{\psi}(\mathcal{G}_{\theta}(\mathbf{z}_q))\|_2^2, \qquad (4)$$

where $\mathbf{z}_q = \mathcal{Q}(\mathcal{E}_{\phi}(\mathbf{x}))$ are quantized tokens. Similar to $\mathcal{L}_{\text{recon}}$, this MSE objective can be enhanced with perceptual loss, e.g., LPIPS (Zhang et al., 2018). During training, we concatenate the AR prediction $\hat{\mathbf{z}}_q = \mathcal{G}_{\theta}(\mathbf{z}_q)$ with $\mathbf{z}_q$ along the batch dimension, and pass them together to the decoder.

The APR loss provides end-to-end generative supervision to the tokenizer directly from the pixel space, and regulates next token prediction loss to be meaningful. This is confirmed by Figure 3a, where end-to-end training with APR loss resolves the latent collapse issue caused by backpropagating the NTP loss only. As shown in Table 1, applying APR loss to the end-to-end training framework improves the overall generation quality, notably lowering the gFID score from 8.01 to 3.32.

### 3.3. Introduce Semantic Representation to Tokenizers

Inspired by the success of representation alignment (Yu et al., 2025c; Leng et al., 2025; Yao & Wang, 2025) in diffusion models, we explore the idea of using semantic features from a vision foundation model $f$ to help improve visual tokenization. To this end, we comprehensively investigate several variants of injecting the representation $f(\mathbf{x})$ into both encoder and decoder of the 1D tokenizer.

**Types of representation injection into encoders.** We first study three methods for injecting semantic representation into the 1D ViT encoder, as illustrated in Figure 4.

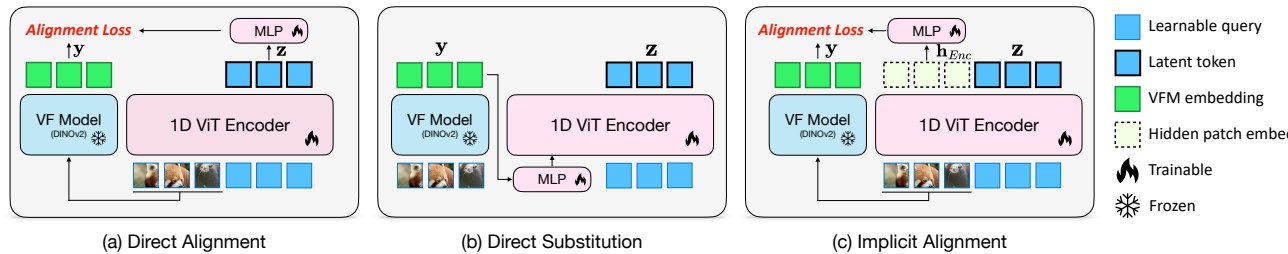

*Figure 4.* **Variants of representation injection into 1D VAE encoder.** Image patches are passed to a vision foundation model (e.g., DINOv2) to extract semantic VFM embeddings. To inject this representation into the 1D ViT encoder, (a) *direct Alignment* means aligning the latent space of tokenizers to the VFM embedding; (b) *direct substitution* replaces 2D image patches with the VFM embedding; and (c) *implicit alignment* aligns the hidden patch embedding to VFM representation.

- **Direct alignment.** Latent tokens $\mathbf{z}$ are directly aligned to pre-trained representation $\mathbf{y} = f(\mathbf{x})$, similar to VA-VAE (Yao & Wang, 2025). Its loss function is defined as

$$\mathcal{L}_{\text{direct}}(\boldsymbol{\omega}, \boldsymbol{\phi}) = -\frac{1}{L}\sum_{\ell=1}^{L}\text{sim}(h_{\boldsymbol{\omega}}(\mathbf{z}^{[\ell]}), \mathcal{I}(\mathbf{y})^{[\ell]}), \quad (5)$$

where $\ell$ is the token index and $\mathcal{I}$ interpolates the 2D features from $\mathbb{R}^{N \times D}$ to $\mathbb{R}^{L \times D}$, and sim measures cosine similarity. This loss enforces the 1D latent codes $\mathbf{z}$ to match the spatially aligned features $f(\mathbf{x})$, which inevitably leaks the 2D spatial prior to the 1D tokenizer.

- **Direct substitution.** An alternative way to leverage visual representation of VFMs is to use them directly as an encoder, which has been explored in training diffusion models (Zheng et al., 2025b; Shi et al., 2025b; Bi et al., 2025; Chen et al., 2025). As demonstrated in Figure 4(b), we replace image patches $\mathbf{x}_{\text{patch}}$ by the projected VFM features, i.e., $\text{MLP}(f(\mathbf{x}))$, concatenate them with learnable quries $\mathbf{q}$, and pass them to the 1D ViT encoder.

- **Implicit alignment.** Instead of directly aligning latent tokens, we consider aligning the hidden patch embeddings $\mathbf{h}_{\text{Enc}}$ to the VFM representations $f(\mathbf{x})$. The 1D latent codes $\mathbf{z}$ are not forced to match the VFM representation, but can still extract semantic information from the aligned 2D hidden embedding. We name this scheme implicit alignment, whose loss function is defined as

$$\mathcal{L}_{\text{implicit}}(\boldsymbol{\omega}, \boldsymbol{\phi}) = -\frac{1}{N}\sum_{n=1}^{N}\text{sim}(h_{\boldsymbol{\omega}}(\mathbf{h}_{\text{Enc}}^{[n]}), \mathbf{y}^{[n]}). \quad (6)$$

**Decoder alignment.** In addition to injecting semantic representation into encoders, we further examine strategies of aligning the decoder to the VFM representation. We hypothesize that the reconstruction task of an 1D ViT decoder is much harder than a 2D ViT decoder, as it requires to recover pixels whose information distributes globally, instead of locally aligns with, the latent token sequence. This is more similar to a conditional generation task, rather than a reconstruction task. Therefore, applying representation alignment

*Table 2.* **Quantitative comparison of injecting representation to 1D and 2D ViT tokenizers.** Numbers are reported by training a EOSTok-L model for 50 epochs. We use 256 query tokens for 1D tokenizer and to match the sequence length of 2D tokenizer for a fair comparison. Images are generated *without* classifier guidance and the prediction accuracy of the AR model is evaluated on the validation dataset. Baseline is trained without VFM representations.

| | rFID ↓ | gFID ↓ | AR Acc. ↑ |
|---|---|---|---|
| ***1D Tokenization*** | | | |
| Baseline | 1.75 | 12.27 | 7.8% |
| + Decoder alignment | 1.12 | 5.68 | 8.2% |
| + (a) Direct alignment | **0.98** | 5.98 | 8.5% |
| (b) Direct substitution | 1.05 | 4.89 | **12.1%** |
| (c) Implicit alignment | 1.02 | **3.32** | 11.9% |
| ***2D Tokenization*** *for reference* | | | |
| Baseline | 1.52 | 12.51 | 5.2% |
| + Decoder & direct alignment | 0.87 | 6.06 | 7.9% |

to 1D ViT decoders could help its convergence, as it does to accelerate diffusion models training. We thus take inspiration from Yu et al. (2025c) and extract the hidden features of the mask tokens from the $k$-th layer of the decoder, denoted as $\mathbf{h}_{\text{Dec}}$, and align it with the VFM features $\mathbf{y} = f(\mathbf{x})$.

**Semantic alignments improve AR generative quality.** We test these approaches of injecting VFM representations on ImageNet generation, comparing them to the baseline without using VFMs. As shown in Table 2, applying *decoder alignment* significantly improves the 1D tokenizer on reconstruction, reflecting on both rFID and gFID. However, the AR prediction accuracy increases marginally, so it does not make AR modeling easier.

For leveraging VFM on 1D ViT encoder, all three approaches slightly improve the reconstruction quality over the baseline. However, applying *direct alignment* worsens the generative quality, which supports our hypothesis that enforcing 2D spatial structures on the latent space is detrimental to the AR modeling. Approaches (b) and (c) abstain from using a 2D spatial prior during training, among

which *implicit alignment* significantly improves generation quality. Moreover, applying both approaches improves the prediction accuracy of the AR generative model by a large margin, indicating that a more AR-generation-friendly tokenization can be learned through semantic guidance during training.

## 4. Experiments

### 4.1. Experimental Setup

**Model Architecture.** As described in Section 3.1, we use a 1D ViT tokenizer with similar architecture as Yu et al. (2025b), and adopt IBQ (Shi et al., 2025a) as the vector quantization module. Unless specified, all models are trained with a $K = 4096$ codebook size, a $L = 256$ token sequence length, and a $d = 64$ latent dimension. We design the attention maps of both the encoder and decoder to be bidirectional over patch tokens but causal over learnable queries. To stabilize training, we apply $\ell_2$-normalization to both the latent vectors before quantization and the IBQ codebook, projecting them to the $K$-dimensional unit sphere. We build our AR generative model based on LlamaGen (Sun et al., 2024), with an additional shared global AdaLN modulation with per-block learnable biases (Chen et al., 2023). We use DINOv2-ViT-L (Oquab et al., 2023) for representation alignment.

**Training.** We train our model on ImageNet-1K 256×256 (Deng et al., 2009) for 400 epochs using Adam optimizer (Kingma & Ba, 2014) with a batch size of 256 and BF16 precision. We train our model on four different sizes (S, B, L, and H), while jointly scaling the tokenizer and the AR generative model. Detailed configurations are provided in Section A.1. We use L2, LPIPS (Zhang et al., 2018), and GAN loss (Goodfellow et al., 2014; Sauer et al., 2023) for reconstruction $\mathcal{L}_{\text{recon}}$, and use L2 and LPIPS for APR loss $\mathcal{L}_{\text{APR}}$. We stabilize the discriminator of GANs by LeCam divergence (Tseng et al., 2021).

**Evaluation.** We use the evaluation code from Dhariwal & Nichol (2021) and report Fréchet Inception Distance (Heusel et al., 2017) (both rFID and gFID), and Inception Score (IS) (Salimans et al., 2016) to evaluate the reconstruction and generation quality. We adopt AutoGuidance (Karras et al., 2024) in replace of classifier-free guidance (Ho & Salimans, 2022).

### 4.2. Generation Results on ImageNet 256×256

We demonstrate the effectiveness of our end-to-end training framework and compare EOSTok with state-of-the-art methods on ImageNet-1K 256×256 benchmark. As shown in Table 3, our EOSTok-L model achieves a 1.74 gFID *without* guidance using only 312M parameters, outperforming the results of competitive 1D tokenization baselines even

with classifier guidance. Moreover, our EOSTok tokenizer has a much better reconstruction quality in terms of rFID, as the EOSTok-L tokenizer achieves an rFID of 0.73 with 165M parameters. This is especially credited to the use of semantic VFM representation alignment on the 1D ViT decoder, which helps the convergence of 1D tokenizer, as shown by the ablation study in Table 2. We also provide some representative images generated by EOSTok in Figure 6.

**Scaling behavior.** We train EOSTok with four different model sizes, from 93M to 644M. As shown in Figure 5a, scaling the AR generative model consistently improves the generation quality as measured by FID. In Figure 5b, we plot the learning curve of the cross-entropy loss, $\mathcal{L}_{\text{NTP}}$, where scaling up the AR model lowers the converging point of the CE loss. Moreover, we jointly scale the ViT tokenizer and the AR model EOSTok to 388M and 644M parameters, respectively, for a total of 1B trainable parameters, enabling the EOSTok-H model to achieve a state-of-the-art gFID of 1.48 without guidance.

### 4.3. Impact of End-to-end Training

**Facilitating AR modeling.** To test the hypothesis that end-to-end training with APR loss builds a more AR generation-friendly latent space, we design an experiment as follows. We take an EOSTok-L tokenizer trained for 50 epochs, freeze it, and then train another AR model from scratch to generate the corresponding discrete visual tokens. We consider three different setups for the AR model:

(a) training on original token sequences;

(b) training on reversed token sequences;

(c) training on randomly (but fixed) ordered sequences.

Without supervising the tokenizer with NTP loss, APR loss, and semantic alignment, the 1D latent space should be agnostic to ordering. However, as shown in Table 4, the model (a) trained on original token sequences has a much better generation FID score compared to both models that are trained with reversed and randomly ordered sequences. This shows that the end-to-end training pipeline back-propagating the NTP and APR losses learns to represent images as token sequences that are much easier to model sequentially for autoregressive models.

*Table 4.* **Training AR models on token sequences with different ordering.** Token sequences are from a tokenizer that is pre-trained using our end-to-end training framework.

|  | Original | Reversed | Random |
|---|---|---|---|
| gFID (w/o guidance) ↓ | **4.10** | 10.27 | 7.81 |
| AR Accuracy ↑ | **10.3%** | 9.5% | 9.8% |

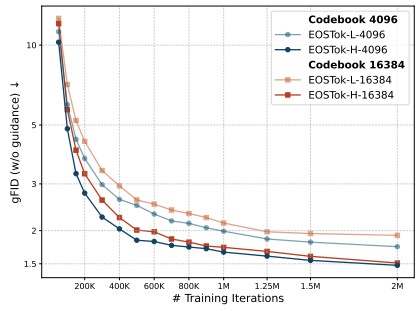

*(a)* gFID vs. training iterations.

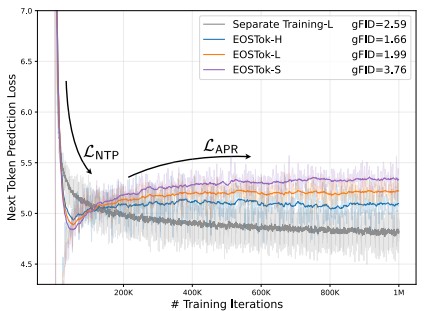

*(b)* Next token prediction loss $\mathcal{L}_{\text{NTP}}$

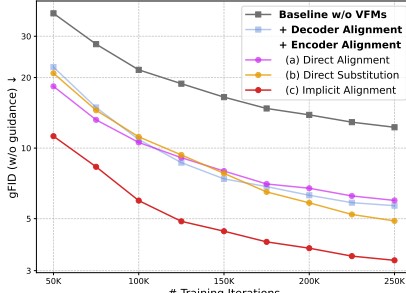

*(c)* The gFID curves of different semantic representation injection schemes.

*Figure 5.* **Learning curves of EOSTok.** (a) We plot the generative FID score evaluated during training. For both codebook sizes $K$ of 4096 and 16384, scaling up the EOSTok model consistently improves the overall generation quality. Moreover, using a larger model size (e.g., EOSTok-H) closes the gap between $K = 4096$ and $K = 16384$. (b) We plot the loss curves for $\mathcal{L}_{\text{NTP}}$ across model sizes (S, L, and H). We compare our end-to-end training with two-stage separate training on model size L. Although separate training achieves lower NTP training loss, our framework achieves better overall generation quality by enabling end-to-end feedback, i.e., $\mathcal{L}_{\text{APR}}$. (c) The convergence of our model using different semantic alignment strategies. Among the three strategies considered for injecting semantic representation into the encoder, *implicit alignment* is the most effective.

*Table 3.* **ImageNet 256 Generation Results.** We categorize existing generative models based on the visual tokenizer used, including **2D continuous** tokenizers with latent diffusion and masked AR models, **2D discrete** tokenizers with AR models and mask models, and **1D** tokenizers with AR models. EOSTok achieves the state-of-the-art performance on generation without using classifier guidance.

| Method | Tokenizer | | | | Generator | | w/o guidance | | w/ guidance | |
| --- | --- | --- | --- | --- | --- | --- | --- | --- | --- | --- |
| | Type | #Params | #Tokens | rFID↓ | Type | #Params | gFID↓ | IS↑ | gFID↓ | IS↑ |
| *2D Continuous Latent Space* | | | | | | | | | | |
| LDM-4 (Rombach et al., 2022) | SD-VAE | 55M | 64×64 | 0.27 | Diff. | 400M | 10.56 | 103.5 | 3.60 | 247.7 |
| DiT-XL/2 (Peebles & Xie, 2023) | SD-VAE | 84M | 32×32 | 0.62 | Diff. | 675M | 9.62 | 121.5 | 2.27 | 278.2 |
| REPA-XL/2 (Yu et al., 2025c) | SD-VAE | 84M | 32×32 | 0.62 | Diff. | 675M | 5.90 | 157.8 | 1.42 | 305.7 |
| Lightning-DiT-XL (Yao & Wang, 2025) | VA-VAE | 84M | 32×32 | 0.28 | Diff. | 675M | 2.17 | 205.6 | 1.35 | 295.3 |
| MAR-L (Li et al., 2024) | SD-VAE | 66M | 16×16 | 0.87 | MAR Diff. | 479M | 2.60 | 221.4 | 1.78 | 296.0 |
| *2D Discrete Tokenization* | | | | | | | | | | |
| VQGAN (Esser et al., 2021) | VQ | 23M | 16×16 | 4.98 | AR | 1.4B | 15.78 | 74.3 | - | - |
| RQTrans (Lee et al., 2022) | RQ | 66M | 8×8×4 | 3.20 | AR | 1.4B | 8.71 | 119.0 | 3.89 | 337.5 |
| DQTrans (Huang et al., 2023) | DQ | 48M | 640 | 4.08 | AR | 655M | 5.11 | 178.2 | - | - |
| MaskGIT (Chang et al., 2022) | VQ | 66M | 16×16 | 2.28 | Mask | 227M | 6.18 | 182.1 | - | - |
| MAGVIT-v2 (Yu et al., 2023) | LFQ | 133M | 16×16 | 1.17 | Mask | 307M | 3.65 | 200.5 | 1.78 | 319.4 |
| LlamaGen-XL (Sun et al., 2024) | VQ | 72M | 16×16 | 0.94 | AR | 775M | 14.77 | 80.8 | 2.62 | 244.1 |
| RAR-L (Yu et al., 2025a) | VQ | 66M | 16×16 | 2.28 | AR | 461M | 5.39 | 149.1 | 1.70 | 299.5 |
| IBQ-L (Shi et al., 2025a) | IBQ | 128M | 16×16 | 1.37 | AR | 649M | - | - | 2.45 | 267.5 |
| VAR-d20 (Tian et al., 2024) | MSRQ | 109M | 680 | 0.90 | VAR | 310M | - | - | 2.57 | 302.6 |
| AliTok-L (Wu et al., 2025) | VQ | - | 16+16×16 | 0.86 | AR | 318M | 1.98 | 200.8 | 1.38 | 326.2 |
| *1D Tokenization* | | | | | | | | | | |
| TiTok-L-32 (Yu et al., 2025b) | 1D VQ | 641M | 32 | 2.21 | Mask | 177M | 3.15 | 173.0 | 2.77 | 199.8 |
| FlexTok d18-18 (Bachmann et al., 2025) | 1D FSQ Flow | 950M | 1-256 | 1.61 | AR | 1.33B | - | - | 2.02 | - |
| Semanticist (Wen et al., 2025) | 1D VAE Flow | - | 1-256 | 0.72 | AR Diff. | 343M | - | - | 2.57 | 254.0 |
| GigaTok (Xiong et al., 2025) | 1D VQ | 622M | 256 | 0.81 | AR | 111M | - | - | 3.26 | 221.0 |
| SpectualAR-d20 (Huang et al., 2025) | 1D VQ | - | 64 | 4.03 | AR | 600M | - | - | 2.49 | 305.4 |
| VFMTok (Zheng et al., 2025a) | 1D VQ | - | 256 | 0.89 | AR | 343M | 2.11 | 230.1 | 2.75 | 278.8 |
| ResTok (Zhang et al., 2026) | 1D VQ | 662M | 128 | 1.28 | HAR | 326M | - | - | 2.34 | 257.8 |
| **EOSTok-S (Ours)** | 1D IBQ | 165M | 256 | 0.74 | AR | 93M | 3.50 | 155.7 | 2.57 | 211.5 |
| **EOSTok-B (Ours)** | 1D IBQ | 165M | 256 | 0.73 | AR | 164M | 2.38 | 185.6 | 1.98 | 220.1 |
| **EOSTok-L (Ours)** | 1D IBQ | 165M | 256 | 0.73 | AR | 312M | 1.74 | 210.2 | **1.35** | 236.5 |
| **EOSTok-H (Ours)** | 1D IBQ | 388M | 256 | **0.71** | AR | 644M | **1.48** | **239.5** | 1.38 | 265.7 |

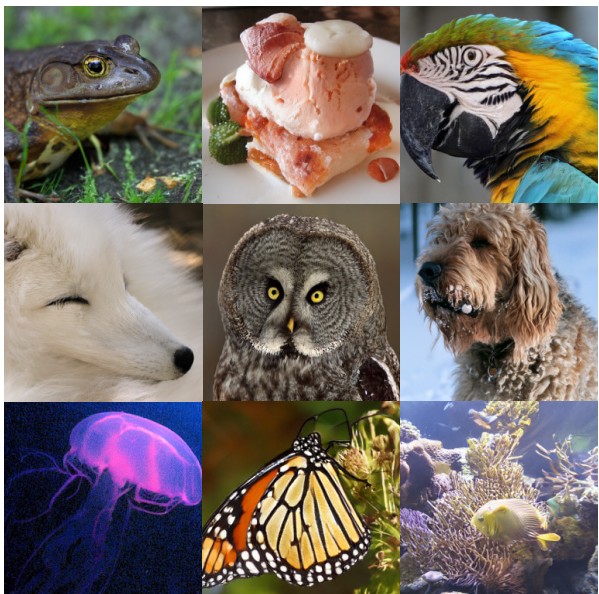

*Figure 6.* **Qualitative results on ImageNet 256 generation.**

**Comparing to two-stage separate training.** We plot the next token prediction (cross entropy) loss of $\mathcal{L}_{\text{NTP}}$ in Figure 5b. Using one-stage end-to-end training with $\mathcal{L}_{\text{APR}}$ results in loss curves that plateau earlier and to a larger value, compared to those of separate training, while the generation FID score is much better than the latter one. This reaffirms that the NTP loss, $\mathcal{L}_{\text{NTP}}$, cannot determine how good the generative model is, and that end-to-end training effectively optimizes for final generation quality.

### 4.4. Ablation Studies

**Sequence length.** Our 1D tokenizer offers the flexibility to compress images into tokens of varying sequence lengths. As shown in Table 6, using a longer sequence length consistently improves the reconstruction quality (i.e., rFID). However, continuing to increase the sequence length makes autoregressive modeling more difficult, and the gFID score peaks at 192, setting up a reconstruction-generation trade-off similar to what was observed in latent channels of latent diffusion models (Yao & Wang, 2025). We also find that nested dropout, introduced in (Bachmann et al., 2025; Wen et al., 2025), is effective for regularizing the token space and expanding the reconstruction-generation frontier, as shown in Section B.

**Codebook size.** We study the effect of choosing different codebook sizes for latent vector quantization. We train a EOSTok-L model for 50 epochs with a codebook size of $K \in [1024, 2048, 4096, 8192, 16384]$. We observe yet another tradeoff between reconstruction and generation. While increasing the number of codes improves the reconstruction FID, it also complicates the next token classification task for AR models.

We conduct additional experiments on training larger mod-

*Table 5.* **Ablation on the choice of sequence length.** Increasing sequence length consistently improves the model's ability for reconstruction but could hurt generation quality if the sequence goes too long.

| Token sequence length | 32 | 64 | 128 | 192 | 256 |
|---|---|---|---|---|---|
| rFID ↓ | 17.50 | 1.94 | 1.32 | 1.08 | **1.02** |
| gFID (w/o guidance) ↓ | 22.37 | 3.18 | 3.09 | **3.04** | 3.32 |

*Table 6.* **Ablation on the choice of codebook size.** Reducing the codebook size trades off reconstruction for generation quality. Our model retains a high code usage when using a larger codebook size.

| Codebook size | 1024 | 2048 | 4096 | 8192 | 16384 |
|---|---|---|---|---|---|
| rFID ↓ | 1.18 | 1.07 | 1.02 | 0.98 | **0.96** |
| gFID (w/o guidance) ↓ | 3.24 | **3.20** | 3.32 | 3.68 | 4.08 |
| Code Usage ↑ | 100% | 100% | 99.7% | 99.7% | 99.2% |

els with codebook sizes of 4096 and 16384. The convergence curves of gFID are shown in Figure 5a. We find that using a larger model (e.g., EOSTok-H) closes the gap between the two codebook sizes as the model converges. Specifically, final gFID scores on EOSTok-L are 1.74 for $K = 4096$ and 1.92 for $K = 16384$, but the results on EOSTok-H are gFID = 1.48 and 1.51, respectively. This indicates that the optimal codebook size $K$ shifts as the AR model's capacity increases, and that scaling up the EOSTok model can alleviate the reconstruction-generation dilemma in choosing codebook size.

**More experimental results.** In Section B, we present additional experimental results, including ImageNet 512 generation results and more reconstruction metrics; and more ablation studies, including choosing different loss weights, dropout strategies, and vision foundation models for representation alignment.

## 5. Conclusion

We present an ***end-to-end*** training pipeline for autoregressive image generation, which jointly optimizes a 1D vision tokenizer and an AR generative model for ***reconstruction, generation, and semantic alignment***. The next token prediction loss of the AR model on discrete latent sequences cannot determine its final generation quality, and we thus propose an ***autoregressive prediction reconstruction*** loss to bridge this gap. We investigate ways to incorporate global semantic information from VFMs into our 1D vision tokenizer, without enforcing the sequential latent space to align with the 2D spatial structure of VFM representations. Built on top of this training pipeline, our EOSTok tokenizer learns a 1D sequential token space that facilitates autoregressive modeling, and significantly improves the overall generation quality of AR models. We instantiate EOSTok across four model sizes, demonstrating a scaling property and achieving a gFID score of 1.48 without guidance.

## Impact Statement

This paper presents work whose goal is to advance the field of Machine Learning. There are many potential societal consequences of our work, none which we feel must be specifically highlighted here.

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

# A. Implementation Details

## A.1. Model Architecture

### A.1.1. TOKENIZER.

Our 1D ViT tokenizer uses a similar architecture as TiTok (Yu et al., 2025b) to compress images into 1D discrete sequences without 2D structural prior.

**Encoder.** Images are first patchified into tokens with a $P \times P$ patch size and hidden dimension of $D$, resulting in a total of $HW/P^2$ patch tokens in a 2D grid. These tokens are flattened and concatenated with $L$ learnable query tokens $\mathbf{q} \in \mathbb{R}^{L \times D}$. We use learnable positional embeddings for simplicity. The embedded tokens of length $HW/P^2 + L$ are then passed into multiple layers of transformer blocks. It outputs $[\mathbf{h}_{\text{Enc}}, \mathbf{z}]$ where $\mathbf{h}_{\text{Enc}} \in \mathbb{R}^{HW/P^2 \times D}$ is the hidden 2D embedding. The query tokens go through a linear projection layer and are mapped to $d$-dimension, i.e., $\mathbf{z} \in \mathbb{R}^{L \times d}$.

**Decoder.** The decoder follows a symmetric design of the encoder. Sequential 1D latent codes are concatenated with $HW/P^2$ mask tokens in a 2D grid. After multiple layers of transformer blocks, the decoder outputs the unmasked tokens, followed by an unpatchify layer and a convolutional output layer.

**Hybrid attention mask.** On our 1D ViT encoder, the attention is bidirectional among 2D patch tokens and causal along the 1D query tokens. Query tokens are allowed to attend to patch tokens, whereas patch tokens cannot attend to query tokens. Similarly, on the decoder, the attention is causal along 1D query tokens and bidirectional among 2D patch tokens.

**Quantizer.** We use IBQ (Shi et al., 2025a) to quantize latent embeddings. We stabilize the codebook training by applying an $\ell_2$ normalization on both the codebook $\mathcal{C}$ and the continuous latent embedding when computing their similarity, i.e.,

$$\text{logits} = \left[ \frac{\mathbf{z}^T \mathcal{C}_1}{\|\mathbf{z}\|_2 \|\mathcal{C}_1\|_2}, \ldots, \frac{\mathbf{z}^T \mathcal{C}_1}{\|\mathbf{z}\|_2 \|\mathcal{C}_K\|_2} \right], \text{ and } \mathbf{p} = \text{softmax}(\text{logits}/\tau). \tag{7}$$

The detailed model configurations for each model size (S, B, L, H) are listed in Table 9.

### A.1.2. AUTOREGRESSIVE GENERATIVE MODEL.

We slightly modify the language model in LlamaGen (Sun et al., 2024) for autoregressive generation. Specifically, we add an additional shared global AdaLN (Peebles & Xie, 2023) modulation with per-block learnable biases. The model applies RMSNorm (Zhang & Sennrich, 2019) and SwiGLU activation function (Shazeer, 2020). Since we tokenize images into 1D sequences, we replace the 2D RoPE with learnable positional embeddings. We include more details on model configurations in Table 9.

## A.2. Training Details

### A.2.1. LOSS FUNCTIONS

Our end-to-end framework jointly trains a tokenizer with encoder $\mathcal{E}_\phi$ and decoder $\mathcal{D}_\psi$, and an AR model $\mathcal{G}_\theta$. The training losses include reconstruction, next-token prediction (NTP), autoregressive prediction reconstruction (APR), and semantic alignment. Putting it together, we summarize the objective function of EOSTok as:

$$\begin{aligned}
\mathcal{L}_{\text{EOSTok}}(\phi, \psi, \theta) = {} & \mathcal{L}_{\text{VQVAE}}(\phi, \psi) + \lambda_{\text{NTP}} \mathcal{L}_{\text{NTP}}(\phi, \theta) + \lambda_{\text{APR}} \mathcal{L}_{\text{APR}}(\phi, \psi, \theta) \\
& + \min_{\omega_1, \omega_2} \lambda_{\text{sem}}(\mathcal{L}_{\text{implicit}}(\omega_1, \phi) + \mathcal{L}_{\text{decoder-align}}(\omega_2, \phi, \psi)).
\end{aligned} \tag{8}$$

$\lambda_{\text{NTP}}, \lambda_{\text{APR}}$ and $\lambda_{\text{sem}}$ are pre-determined loss weights, and $\omega_1, \omega_2$ are learnable MLP layers for representation alignment. The VAE loss can be decomposed into $\mathcal{L}_{\text{VQVAE}} = L_2 + \text{LPIPS} + \lambda_{\text{GAN}} \mathcal{L}_{\text{GAN}} + \lambda_{\text{reg}} \mathcal{L}_{\text{reg}}$, and $\mathcal{L}_{\text{APR}} = L_2 + \text{LPIPS}$. Furthermore, we use the discriminator from StyleGAN-T (Sauer et al., 2023) for GAN loss, with LeCAM divergence (Tseng

*Table 7.* **Computation and memory cost.**

|                                                      | EOSTok-L   | EOSTok-H   |
| ---------------------------------------------------- | ---------- | ---------- |
| Tokenizer GFLOPs (Encoder + Decoder)                 | 91 + 91    | 210 + 210  |
| AR Transformer GFLOPs                                | 162        | 338        |
| VFM embedder GFLOPs                                  | 162        | 162        |
| Forward total (AR + Encode + 2× Decode + VFM)        | 597        | 1130       |
| Overhead compared to two-stage training              | 15.2%      | 18.6%      |
| Peak memory (Per device batch size 32)               | 36.51 GB   | 56.96 GB   |

*Table 8.* **Sampling efficiency of EOSTok.** (*The original paper uses a different convention of 1 mul-add = 1 op, which halves the number.)

| EOSTok-H | AR Model (256 tokens) | Decoder | Total |
| -------- | --------------------- | ------- | ----- |
| GFLOPs   | 342.3                 | 210.1   | 552.4 |
| DiT-XL/2 | Diffusion Model       | Decoder | Total |
| GFLOPs*  | 237.2 × 250 steps     | 622.2   | 59.9k |

et al., 2021) to stabilize training. The perceptual LPIPS loss is computed with a VGG (Simonyan & Zisserman, 2014) backbone. We include details of loss weights in Table 9.

### A.2.2. OPTIMIZER

We use Adam optimizers with an initial learning rate of 1e-4 for both the tokenizer and the AR model, and employ a cosine learning rate scheduler that decays to 1e-6 at 2M iterations. For the discriminator, we use an Adam optimizer with a fixed learning rate of 1e-4. We train our models on 8 H100 GPUs with a batch size of 256 for 400 epochs (approximately 2M iterations). More training details can be found in Table 9.

### A.3. Sampling Details

We apply autoregressive sampling with KV cache, using a temperature of 1.0 without top-k or top-p strategies. We use classifier guidance on EOSTok-S and EOSTok-B, i.e., $\ell_g = \ell_u + s(\ell_c - \ell_u)$, where $\ell_u$ and $\ell_c$ are logits of unconditional and class conditional sampling, and $s$ is the CFG scale. We notice a diminishing effect on applying CFG sampling when scaling up our model to EOSTok-L and EOSTok-H as the unconditional generation quality of the model improves. We thus apply AutoGuidance following (Karras et al., 2024) by training a lightweight AR model on the tokenizer and using it to replace the unconditional logits. We use the same budget for all models to search for the best guidance scale.

### A.4. Computation Cost

We provide a detailed cost analysis of EOSTok in Table 7, including the GFLOPs of each module and the peak memory usage in training. Moreover, the sampling GFLOPs of our largest model EOSTok-H are listed in Table 8. Since we use KV cache during sampling, the sampling speed is much faster than diffusion sampling. With batched image generation, EOSTok-H can generate about 10.5 images per second on a single H100. As shown in Table 8, EOSTok-H is 20 to 100 times faster than the DiT-XL/2 model, depending on the diffusion sampling algorithm used.

*Table 9.* **Implementation details of EOSTok on ImageNet 256 generation experiments.**

|  | EOSTok-S | EOSTok-B | EOSTok-L | EOSTok-H |
|---|---|---|---|---|
| ***Tokenizer configs*** | | | | |
| Params. | 165M | 165M | 165M | 388M |
| Patch size | 16 | 16 | 16 | 16 |
| Transformer layers (encoder-decoder) | 12-12 | 12-12 | 12-12 | 16-16 |
| Hidden dimensions | 768 | 768 | 768 | 1024 |
| Attention heads | 12 | 12 | 12 | 16 |
| Latent space dimensions | 64 | 64 | 64 | 64 |
| ***Quantizer*** | | | | |
| Codebook size | 4096 | 4096 | 4096 | 4096 |
| Temperature | 1.0 | 1.0 | 1.0 | 1.0 |
| Regularization weights $\lambda_{reg}$ | 1e-3 | 1e-3 | 1e-3 | 1e-3 |
| Entropy weights | 0.01 | 0.01 | 0.01 | 0.01 |
| ***Autoregressive Model configs*** | | | | |
| Params. | 93M | 164M | 312M | 644M |
| Layers | 12 | 12 | 24 | 32 |
| Hidden dimensions | 768 | 1024 | 1024 | 1280 |
| Attention heads | 12 | 16 | 16 | 20 |
| ***Training loss weights*** | | | | |
| Reconstruction L2 | 1.0 | 1.0 | 1.0 | 1.0 |
| Reconstruction LPIPS | 1.0 | 1.0 | 1.0 | 1.0 |
| Implicit Alignment | 1.0 | 1.0 | 1.0 | 1.0 |
| GAN loss | 0.1 | 0.1 | 0.1 | 0.1 |
| GAN LeCam | 0.05 | 0.05 | 0.05 | 0.05 |
| APR L2 | 1.0 | 1.0 | 1.0 | 1.0 |
| APR LPIPS | 1.0 | 1.0 | 1.0 | 1.0 |
| Next token prediction | 0.1 | 0.1 | 0.1 | 0.01 |
| ***Training details*** | | | | |
| Batch size | 256 | 256 | 256 | 256 |
| Epochs | 400 | 400 | 400 | 400 |
| Learning rate | 1e-4 | 1e-4 | 1e-4 | 1e-4 |
| Cosine learning rate min | 1e-6 | 1e-6 | 1e-6 | 1e-6 |
| Adam $\beta_1$ | 0.9 | 0.9 | 0.9 | 0.9 |
| Adam $\beta_2$ Tokenizer/AR | 0.999/0.95 | 0.999/0.95 | 0.999/0.95 | 0.999/0.95 |
| EMA decay rate | 0.9999 | 0.9999 | 0.9999 | 0.9999 |
| Nested dropout ratio | 0.5 | 0.5 | 0.5 | 1.0 |
| Class dropout ratio | 0.1 | 0.1 | 0.1 | 0.1 |

## B. Additional Experimental Results

**APR weights.** We further study the effect of using different APR weights in end-to-end training. We fix the NTP loss weight $\lambda_{\text{NTP}}$ to 0.1 in this experiment, and train an EOSTok-L model for 50 epochs. As shown in Table 10, using an APR weight of 1.0 achieves the best performance for both reconstruction and generation quality.

**Nested dropout.** We conduct experiments to understand the effect of applying nested dropout proposed in (Bachmann et al., 2025; Wen et al., 2025) to regularize the 1D sequential latent space and compress important information into tokens in the front. We consider $p = [0.0, 0.25, 0.5, 1.0]$ for the probability of applying nested dropout during training. As shown in Table 11, using a nested dropout strategy can further make latent tokens easier for AR modeling. Specifically, applying nested dropout with probability $p = 1$ results in a 17.6% AR prediction accuracy, while it slightly worsens generation quality since the over-compressing hurts the model's ability of reconstructing images. We thus apply nested dropout with rate $p = 0.5$ in our training pipeline.

Table 10. **Ablation on the APR loss weight.**

| APR loss weight $\lambda_{\text{APR}}$ | 0.0 | 0.5 | 1.0 | 2.0 | 4.0 |
|---|---|---|---|---|---|
| rFID ↓ | 1.03 | **1.02** | **1.02** | 1.05 | 1.12 |
| gFID (w/o guidance) ↓ | 4.09 | 3.52 | **3.32** | 3.34 | 3.57 |

Table 11. **Ablation on the nested dropout rate.**

| Dropout probability | 0.0 | 0.25 | 0.5 | 1.0 |
|---|---|---|---|---|
| rFID ↓ | **0.85** | 0.94 | 1.02 | 1.24 |
| gFID (w/o guidance) ↓ | 3.70 | 3.52 | **3.32** | 3.50 |
| AR Acc. ↑ | 10.2 % | 10.8% | 11.9% | **17.6%** |

**The choice of VFM in representation alignment.** We further experiment with EOSTok-L in our ablation setting using a SigLIP2 (Tschannen et al., 2025) model, which contains richer global semantic information. As shown in Table 12, SigLIP2 slightly improves the generative results compared to DINOv2, which shows the robustness of our framework to the choice of pretrained vision foundation models.

Table 12. **Ablation on the choice of vision foundation models for representation alignment.**

| Pretrained VFM | DINOV2 | SigLIP2 |
|---|---|---|
| rFID | 1.02 | 0.88 |
| gFID (w/o guidance) | 3.32 | 3.02 |

Table 14. **More reconstruction metrics of the EOSTok tokenizer.**

| Models | PSNR | SSIM | LPIPS | rFID |
|---|---|---|---|---|
| EOSTok-L (n=4096) | 22.15 | 0.67 | 0.231 | 0.73 |
| IBQ (n=16384) | 22.01 | 0.61 | 0.224 | 1.37 |
| IBQ (n=262144) | 22.69 | 0.64 | 0.203 | 1.00 |
| GigaTok-B-L (n=16384) | 21.21 | 0.68 | 0.206 | 0.81 |
| LlamaGen (n=16384) | 20.79 | 0.62 | - | 2.19 |
| Semanticist (continuous 1D) | 21.61 | 0.63 | - | 0.78 |

Table 13. **Scalability of EOSTok to ImageNet 512.**

| Models | gFID (w/o guidance) |
|---|---|
| DiT-XL/2 (Peebles & Xie, 2023) | 12.03 |
| MaskDiT (Zheng et al., 2023) | 10.79 |
| TiTok-B-128 (Yu et al., 2025b) | 4.17 |
| TiTok-L-64 (Yu et al., 2025b) | 3.99 |
| **EOSTok-L (Ours)** | **1.98** |

**Scalability to higher resolution.** We show that EOSTok can be naturally extended to higher resolution. We uses the same architecture of EOSTok-L, keeping the patch size of 16 and the AR sequence length of 256. As shown in Table 13, our EOSTok-L achieves an gFID of 1.98 without guidance, outperforming existing diffusion models with 2D tokenizers, and also existing mask based generative models with 1D tokenizers.

**Reconstruction evaluations.** We include more comprehensive reconstruction metrics in Table 14. We include the results of recent discrete tokenizers for reference. Our tokenizer achieves comparable performance on PSNR, SSIM, and LPIPS despite using compact 1D compression and better distributional performance, as measured by rFID.

