# OpenReview forum: "End-to-End Autoregressive Image Generation with 1D Semantic Tokenizer"
_ICML.cc/2026/Conference — ICML 2026 spotlight_

### Official Review · Reviewer_fFvJ · 2026-03-05

**Soundness:** 4
**Presentation:** 4
**Significance:** 4
**Originality:** 3
**Overall Recommendation:** 5
**Confidence:** 4

**Summary:**

This paper addresses a clear limitation in current autoregressive image generation pipelines: the tokenizer is typically trained only for reconstruction and then frozen, even though the final objective is generation. The authors argue that next-token prediction accuracy alone is not a sufficient signal for image quality, and they demonstrate this gap empirically. They propose a fully end-to-end framework that jointly trains a 1D discrete tokenizer and an autoregressive transformer. The 1D latent representation is designed to be naturally compatible with causal modeling. Beyond standard reconstruction and next-token losses, they introduce an autoregressive prediction reconstruction (APR) loss, which decodes predicted tokens back to pixel space to provide direct supervision on generation quality. They further incorporate semantic signals from a vision foundation model through alignment strategies that improve the learned representation without reintroducing a 2D spatial prior. The resulting formulation is conceptually clean and well-motivated. Empirically, the method achieves state-of-the-art ImageNet 256×256 generation performance without guidance and shows consistent improvements with scale.

**Compliance With Llm Reviewing Policy:**

Affirmed.

**Final Justification:**

The rebuttal confirmed my initial assessment of the paper.

**Key Questions For Authors:**

- The paper focuses on the benefits of end-to-end training and APR loss. Could the authors elaborate on the main practical downsides or difficulties of the approach? For example, are there stability issues, hyperparameter sensitivities, or additional computational costs compared to two-stage training?
- Could the authors provide details on generation time and sampling efficiency? In particular, how does inference latency scale with the number of latent tokens, and how does it compare to strong diffusion baselines?

**Limitations:**

No, see questions.

**Strengths And Weaknesses:**

- Soundness: The paper is technically solid. The proposed end-to-end training framework is clearly defined, and each component (joint training, APR loss, semantic alignment) is motivated and evaluated through ablations. In particular, the analysis of the mismatch between next-token prediction loss and pixel-level generation quality is convincing and supported by controlled experiments. The ablation studies on sequence length, codebook size, alignment strategies, and scaling provide strong empirical backing for the claims. The experimental setup is standard and appropriate for ImageNet 256×256 generation, and comparisons against strong baselines are comprehensive. Overall, the claims are well supported by empirical evidence, and the methodology appears careful and consistent.
- Presentation: The paper is clearly written and logically structured. The motivation is easy to follow, and the progression from identifying the problem (two-stage mismatch) to proposing end-to-end training with APR and semantic alignment is coherent. Figures and tables are helpful, especially those illustrating latent collapse and scaling behavior.
- Significance: The paper addresses the meaningful problem in autoregressive image generation of designing tokenizers that are genuinely optimized for downstream generation rather than only reconstruction. This is a relevant and timely direction given renewed interest in AR visual models as alternatives to diffusion. Achieving state-of-the-art ImageNet 256×256 results without guidance is a strong empirical result. Beyond benchmark performance, the end-to-end training principle and the APR loss could influence future tokenizer and AR model design.
- Originality: The introduction of the APR loss to explicitly connect token-level prediction with pixel-space reconstruction is a simple yet insightful contribution. It directly addresses the mismatch between next-token prediction accuracy and actual image quality, which is often overlooked. The end-to-end joint optimization of a 1D tokenizer and an autoregressive model represents a meaningful conceptual shift from standard two-stage training. In addition, the use of vision foundation model (VFM) alignment adds a well justified semantic regularization component.

---

> ### Author Rebuttal · Authors · 2026-03-31
>
> We thank the reviewer for the positive and encouraging evaluation, and for the excellent scores on soundness, presentation, and significance. We are happy to provide additional details below.
>
> 1. **Practical difficulties of end-to-end training and APR loss.** A major downside of end-to-end training is training instability, as illustrated in Figure 2 and Table 1. The APR loss largely alleviates this issue by binding pixel-space generative supervision to the NTP loss. To further stabilize training, we use a 20k-step learning rate warmup schedule for both the tokenizer and AR model parameters. With APR loss and learning rate warmup applied, we found the training process to be stable and robust to loss weight hyperparameters. Another downside of the APR loss is that it introduces additional computation overhead from decoding the AR predictions, adding 18.6% more GFLOPs in training for our largest model. This overhead is partially mitigated by parallelizing the decoding of both the latent codes and AR predictions during training.
>
> 2. **Sampling efficiency.** We list the sampling GFLOPs for our largest model EOSTok-H in the table below. Since we use KV cache during sampling, the sampling speed is much faster than diffusion sampling. With batched image generation, EOSTok-H can generate about 10.5 images per second on a single H100. As shown in the comparison below, EOSTok-H is 20--100 times faster than the DiT-XL/2 model, depending on the diffusion sampling algorithm used.
>
>    | **EOSTok-H** | **AR model (256 tokens)** | **Decoder** | **Total** |
>    | ------------ | ------------------------- | ----------- | --------- |
>    | GFLOPs       | 342.3                     | 210.1       | 552.4     |
>    | **DiT-XL/2** | **Diffusion model**       | **Decoder** | **Total** |
>    | GFLOPs*      | 237.2 x 250 steps         | 622.2       | 59.9K     |
>
>    *The original paper uses a different convention of 1 mul-add = 1 op, which halves the number.
>
>    We also report how the inference time of the AR model scales with the number of latent tokens for different sequence lengths. The sampling cost is dominated by linear layers in the AR model rather than the attention layers, making it scale more linearly than quadratically.
>
> | EOSTok-H              | 64 tokens | 256 tokens | 1024 tokens |
> | --------------------- | --------- | ---------- | ----------- |
> | Total sampling GFLOPs | 297.0     | 552.4      | 2,082.6     |

---

> > ### Author Rebuttal · Reviewer_fFvJ · 2026-04-02
> >
> > I thank the authors for their reply. I am happy to confirm my score, and I would like to see the discussion on limitations incorporated into the revised manuscript.

---

> > > ### Author Response · Authors · 2026-04-08
> > >
> > > We thank the reviewer for the positive feedback. Following the discussion with other reviewers, we are actively extending EOSTok to ImageNet 512 and would like to further provide some intermediate results. While our training recipe runs for 400 epochs in total, we report generation metrics at 20 and 40 epochs here due to the rebuttal phase time limit. As shown in the table below, EOSTok-L achieves 2.65 gFID without guidance at epoch 40, outperforming all baselines with many more training epochs. We will include the discussion of limitations and the additional experimental results in our revised version.
> > >
> > > | ImageNet 512 | Epoch | gFID (w/o guidance) |
> > > | ------------ | ----- | ------------------- |
> > > | EOSTok-L     | 20    | 3.99                |
> > > | **EOSTok-L** | **40**    | **2.65**            |
> > > | TiTok-L-64   | 200   | 3.91                |
> > > | TiTok-B-128  | 200   | 4.17                |
> > > | DiT-XL/2     | 600   | 12.03               |
> > > | MaskDiT      | 800   | 10.79               |

---

### Official Review · Reviewer_m9oV · 2026-03-09

**Soundness:** 3
**Presentation:** 3
**Significance:** 3
**Originality:** 3
**Overall Recommendation:** 4
**Confidence:** 4

**Summary:**

This paper proposes EOSTok, an end-to-end framework for autoregressive image generation based on a one-dimensional visual tokenizer jointly trained with an autoregressive generator. The work is motivated by two observations: (1) a structural mismatch between conventional 2D grid tokenizers and the sequential factorization required by autoregressive modeling, and (2) the discrepancy between the next-token prediction (NTP) objective and the final pixel-space generation quality.

To address these issues, the authors depart from the standard two-stage training paradigm in which the tokenizer is trained for reconstruction and subsequently frozen while training the autoregressive model. Instead, the tokenizer and AR generator are trained jointly in a single end-to-end pipeline. To prevent latent collapse caused by the NTP objective, the paper introduces an Autoregressive Prediction Reconstruction (APR) loss, which decodes teacher-forced AR predictions back to pixel space and supervises them with reconstruction losses.

The framework further incorporates semantic representation alignment using a frozen vision foundation model (DINOv2). Rather than directly aligning the 1D latent tokens with spatially structured features, the authors propose an implicit alignment strategy that aligns intermediate encoder representations with VFM embeddings.

Experiments on ImageNet-1K 256×256 demonstrate strong empirical performance. The largest model (EOSTok-H) achieves a gFID of 1.48 without classifier guidance, which is reported as state-of-the-art among autoregressive image generation approaches using discrete tokenization.

**Compliance With Llm Reviewing Policy:**

Affirmed.

**Final Justification:**

The authors provided detailed stability analysis, training-cost comparisons, and strong preliminary ImageNet-512 results during the rebuttal, which substantially alleviated my concerns about the reliability and scalability of the proposed end-to-end framework, although some questions about full large-scale validation remain

**Key Questions For Authors:**

1.What is the computational cost of the end-to-end training pipeline compared to the conventional two-stage tokenizer + AR training pipeline (in terms of GPU hours and peak memory)?

2.The APR loss operates on teacher-forced predictions. Have the authors experimented with decoding multi-step autoregressive predictions during training to better approximate inference-time generation?

3.During early training stages, the tokenizer codebook and AR targets are essentially random. How does the AR model maintain stability under such conditions? Were any curriculum strategies or partial freezing schedules used?

4.Since semantic alignment with DINOv2 emphasizes high-level representations, does this affect the tokenizer’s ability to represent high-frequency visual patterns such as text or fine textures?

**Limitations:**

No; the initial submission did not sufficiently discuss the limitations, and I encourage the authors to follow through on their rebuttal commitments by adding a candid discussion in the final version on the challenges of scaling to higher resolutions and the computational overhead of end-to-end training.

**Strengths And Weaknesses:**

Soundness
Strengths：The empirical results are strong and clearly demonstrate the effectiveness of the proposed training framework. On ImageNet-256 generation, the EOSTok-H model achieves a gFID of 1.48 without guidance, outperforming prior autoregressive approaches based on discrete tokenization.

The paper also provides convincing diagnostic experiments illustrating the limitations of the next-token prediction objective when applied to discrete visual tokens. In particular, the authors show that optimizing only the NTP loss leads to codebook collapse, where the tokenizer degenerately uses only a small subset of tokens. The proposed APR loss mitigates this issue by introducing pixel-level supervision on the autoregressive predictions, which stabilizes codebook utilization and improves generation quality.

Weaknesses：While empirically effective, the theoretical justification of the APR loss remains somewhat limited. The reconstruction loss is computed only on single-step teacher-forced predictions, whereas inference-time autoregressive generation involves long chains of sequential predictions. As a result, it is unclear whether the decoder learns a more generative-friendly latent space, or whether it primarily learns to correct one-step prediction artifacts during training.

Another concern is training stability. Jointly optimizing a discrete tokenizer and an autoregressive transformer via straight-through estimation is known to be unstable in many discrete representation learning setups. Although the paper reports successful training, it does not provide sufficient analysis of early-stage training dynamics when the codebook and AR targets are effectively random.

Presentation
Strengths：The manuscript is generally well structured and clearly explains the design of the end-to-end training pipeline. The ablation studies are reasonably comprehensive and provide useful insights into the contributions of different components, including APR loss and semantic representation alignment.

The visualization of codebook utilization and latent representations provides intuitive evidence for the latent collapse issue and the benefits of the proposed training strategy.

Weaknesses：Several implementation details that are critical for reproducibility are insufficiently described. In particular, jointly training a tokenizer and a large autoregressive model requires careful balancing of gradients, loss weights, and learning rate schedules. The paper briefly lists loss weights and architectural configurations but does not sufficiently discuss how training stability was achieved in practice.

Given the scale of the joint optimization problem, additional discussion of training procedures and stabilization strategies would improve reproducibility.

Significance
The work addresses an important problem in autoregressive visual generation: designing tokenizers that are better aligned with sequential modeling. The strong ImageNet generation results demonstrate that 1D tokenization combined with end-to-end generative supervision can be highly effective.

However, the proposed training paradigm sacrifices a key advantage of tokenization-based pipelines—modularity. In conventional approaches, tokenizers can be reused across different generative models or downstream tasks. In contrast, the proposed framework tightly couples the tokenizer with a specific autoregressive generator, meaning that scaling the generator architecture or model size may require retraining the entire tokenizer from scratch. This significantly increases the computational cost and may limit practical adoption.

Originality
The overall idea of jointly training tokenizers and generative models is interesting and well motivated. The APR loss provides a straightforward mechanism for introducing pixel-level generative supervision into the tokenizer training process.

However, most architectural components of the system are derived from existing work. The tokenizer architecture follows closely the design of TiTok-style 1D tokenizers, the quantization mechanism relies on IBQ quantization, and semantic representation alignment with vision foundation models has been explored in several recent works on generative modeling.

As a result, the primary novelty lies in the system-level integration of these components under an end-to-end training pipeline, rather than the introduction of fundamentally new algorithmic ideas.

---

> ### Author Rebuttal · Authors · 2026-03-31
>
> We thank the reviewer for the thorough and insightful review. We appreciate the careful analysis of both the strengths and weaknesses of our work, and we address each concern below.
>
> 1. **Computation and memory cost.** Our EOSTok training scheme combines conventional two-stage training into one end-to-end training stage (Figure 1). The total GPU hours are roughly the same as a two-stage tokenizer + AR training pipeline. The only additional training overhead is the APR loss computation, which costs one extra VAE decoding per step. This decoding is also parallelized with the original VAE decoding for reconstruction loss.
>
>    We further provide a detailed cost analysis of EOSTok, including GFLOPs of each module and the peak memory usage during training. We use 8 H100 GPUs for training, which takes up to 10 days for the largest EOSTok model with a batch size of 32 per device (256 globally). We will make this clear in our revision.
>
> || EOSTok-L              | EOSTok-H                |
> |-|-|-|
> | Tokenizer GFLOPs                                    | 91 Encode + 91 Decode | 210 Encode + 210 Decode |
> | AR generator GFLOPs                                 | 162                   | 338                     |
> | VFM embedder GFLOPs                                 | 162                   | 162                     |
> | **Forward total (AR + Encode + 2x Decode + VFM)**   | **597**               | **1130**                |
> | **Overhead compared to two-stage training**         | **15.2%**             | **18.6%**               |
> | **Training Peak memory (batch size per device=32)** | **36.51 GB**          | **56.96 GB**            |
>
> 2. **APR loss with multi-step AR prediction.** Thanks for the suggestion. We use teacher forcing for the APR loss in EOSTok because it requires only one forward pass and the only computation overhead is one VAE decoding step. We find that the APR loss with teacher forcing is sufficient for providing end-to-end generative supervision at the pixel level. It would be an interesting future direction to explore self-forcing using multi-step autoregressive prediction as a finetuning stage.
>
>     We also want to clarify regarding the comment: *it is unclear whether the decoder learns a more generative-friendly latent space, or whether it primarily learns to correct one-step prediction artifacts during training.* This is an important question that we also consider critical to study. We designed an experiment in Section 4.3 to validate our claim that the tokenizer indeed learns a more generative-friendly latent space. We take the end-to-end trained tokenizer and retrain another AR model from scratch, using three different orders: original, reversed, and random. As shown in Table 4, the AR model trained using the original order achieves a much better gFID compared to the other two orders, demonstrating that our end-to-end training indeed learns a latent space that is more friendly to autoregressive modeling.
>
> 3. **Training stability concerns of joint training.** For training stability, we apply a short linear learning rate warmup of 20k steps, which constitutes 1% of the total training steps. We unfreeze all parameters from the beginning and do not use any specific strategies beyond standard warmup to stabilize training. We provide all hyperparameters of training losses and the optimizer in Table 7 of Appendix A.2. We do not ablate or tune the loss weights except for the APR loss weight, where we find that using a weight of 1 is sufficient for good performance.
>
>     In fact, stabilizing joint training is one of the main goals of our APR loss. As shown in Table 1 and Figure 3, without the APR loss, joint training is indeed unstable due to the gap between NTP loss and final generative quality. However, this instability is resolved by providing direct pixel-space supervision through the APR loss.
>
> 4. **DINOv2 alignment.** Our EOSTok applies implicit alignment to DINOv2 (Figure 4), which only aligns the 2D hidden representations rather than the 1D latent tokens to the pretrained VFM. This design does not enforce the latent space to be strictly aligned, and the ability to represent high-frequency details relies on the reconstruction losses (L2, LPIPS, and GAN).
>
>     To validate this claim, we further experiment with EOSTok-L in our ablation setting using the more recent SigLIP2 [1] model, which contains richer global semantic information compared to DINOv2 that focuses on patch-level semantic features. As shown in the table below, using SigLIP2 slightly improves generative results compared to DINOv2. This suggests that EOSTok benefits from implicit alignment to VFMs, and can gain even more from aligning with high-level global semantic information.
>
> || EOSTok-L with DINOv2|EOSTok-L with SigLIP2|
> |-|-|-|
> |gFID (w/o guidance)|3.32|3.02|
> |rFID|1.02|0.88|
>
> [1] Tschannen et al. "SigLIP 2: Multilingual Vision-Language Encoders with Improved Semantic Understanding, Localization, and Dense Features." arXiv: 2502.14786.

---

> > ### Author Rebuttal · Reviewer_m9oV · 2026-04-04
> >
> > After considering the rebuttal, I believe the paper is technically solid: the added evidence on training cost, stability, and VFM robustness resolves several of my original concerns, although the lack of direct validation beyond ImageNet-256 and the still-limited evidence on scalability remain unaddressed.

---

> > > ### Author Response · Authors · 2026-04-08
> > >
> > > We thank the reviewer for the constructive feedback and prompt response.
> > >
> > > We are actively running EOSTok-L on ImageNet 512 and would like to further provide some intermediate results. While our training recipe runs for 400 epochs in total, we report generation metrics at 20 and 40 epochs here due to the rebuttal phase time limit. We compare it with TiTok, the only 1D tokenizer paper in Table 3 with ImageNet 512 results, as well as diffusion and mask AR baselines. As shown in the table below, EOSTok-L achieves 2.65 gFID without guidance at epoch 40, outperforming all baselines with many more training epochs. We will include full experimental results in the revised version.
> > >
> > > | ImageNet 512 | Epoch  | gFID (w/o guidance) |
> > > | ------------ | ------ | ------------------- |
> > > | EOSTok-L     | 20     | 3.99                |
> > > | **EOSTok-L** | **40** | **2.65**            |
> > > | TiTok-L-64   | 200    | 3.91                |
> > > | TiTok-B-128  | 200    | 4.17                |
> > > | DiT-XL/2     | 600    | 12.03               |
> > > | MaskDiT      | 800    | 10.79               |

---

### Official Review · Reviewer_zBUU · 2026-03-10

**Soundness:** 3
**Presentation:** 3
**Significance:** 3
**Originality:** 3
**Overall Recommendation:** 5
**Confidence:** 4

**Summary:**

In this work, the authors considered the problems: 1) AR image generation is always trained with two stages: vision encoder / decoder and AR model. 2) The training losses / targets in the two stages are not aligned. Meanwhile, the visual encoder / decoder actually always ignores the causal inference for AR models. To this end, the authors considered end-to-end training and introduced a design for an Autoregressive Prediction Reconstruction (APR) loss and proposed EOSTok. Experimental results indicated strong performance.

**Compliance With Llm Reviewing Policy:**

Affirmed.

**Ethical Review Concerns:**

No Ethical Review Concerns

**Final Justification:**

Thanks for the additional experiments and results. I particularly appreciate the reconstruction results, which actually validate the reasonable performance of the proposed vision encoder.

**Key Questions For Authors:**

1. As shown in Fig. 2, what is the necessity for the 1D Causal ViT Decoder? As I understand, we can decode all tokens together, resulting in bi-directional interaction among predicted tokens. No need for a 1D causal decoder, which may harm the reconstruction performance.

2. Could the authors provide some insights for Fig. 3 (a)? What makes it achieve a relatively more uniform distribution? Or why can autoregressive prediction reconstruction (APR) loss help?

3. I would like to see if the 1D causal tokenizer can be utilized standalone. If not, why? If yes, could the authors report the reconstruction performance?

4. While the idea is interesting, I do notice that the experiment is only conducted on ImageNet 256, not even 512.

**Limitations:**

This work presented Impact Statement, while maybe no enough limitations discussed.

**Strengths And Weaknesses:**

**Soundness**: The work deals with the most important problems in the field of AR image generation, the motivation is clear, and the solution is smart. I do think the work is reasonable and good.

**Presentation**: The presentation of this submission is great, clear, and easy to follow.

**Significance**: This work is of great interest and may have a good impact on the following works in this field.

**Originality**: I like the studies and solutions in this work. It introduced great novelty and is not incremental work.

-----
In general, I like this work, and would like to have a deep study.

---

> ### Author Rebuttal · Authors · 2026-03-31
>
> We sincerely thank the reviewer for the positive evaluation and thoughtful questions. We are glad that the reviewer appreciates the novelty and soundness of our work. We address each question below.
>
> 1. **1D causal ViT decoder.** We use causal attention for 1D ViT encoders and decoders in EOSTok because it encourages the tokenizer to learn a more autoregressive-friendly latent space, as suggested in previous works [1,2]. However, the causal attention design is not strictly necessary, and our ablation study suggests that it does not significantly affect performance. As shown in the table below, switching from a bidirectional attention mask to a causal one slightly sacrifices reconstruction quality in favor of generation quality. It is also possible to finetune the decoder as a bidirectional ViT to further enhance reconstruction results, but we did not do so for simplicity.
>
>    |                     | Causal | Bidirectional |
>    | ------------------- | ------ | ------------- |
>    | gFID (w/o guidance) | 3.32   | 3.53          |
>    | rFID                | 1.02   | 0.96          |
>
> 2. **More insights on APR loss.** Figure 3(a) shows that naively backpropagating the next-token prediction loss to the tokenizer collapses the latent space. This is because when the latent space is not fixed, the NTP loss favors a more degenerate latent representation. In the extreme case, when the tokenizer outputs sequences of identical tokens, the NTP loss can easily reach zero. This means the tokenizer can exploit the NTP loss by collapsing the latent representation.
>
>    By introducing the APR loss, we require the next-token predictions to match the ground truth image after decoding. This loss design prevents the tokenizer from exploiting the NTP objective, and therefore alleviates the collapse issue as shown in Figure 3. We will make this clearer in Section 3.2.
>
> 3. *Can the 1D causal tokenizer be utilized standalone?* Yes, the 1D tokenizer can be used standalone for reconstruction. We report the reconstruction FID in Table 3, with rFID 0.73 for EOSTok-L and 0.71 for EOSTok-H. Since our tokenizer is quantized and highly compressed for AR generation, the reconstruction quality is lower than that of continuous 2D embeddings (e.g., 0.62 from SD-VAE and 0.28 from VA-VAE), but it still achieves better reconstruction FID compared to other discrete tokenizers.
>
> 4. *Experiments are conducted on ImageNet 256.* In our paper, we benchmark EOSTok models on ImageNet 256 because most baseline methods in Table 3 only report generation results on ImageNet 256, but in principle EOSTok can be directly applied to ImageNet 512 training as well.
>
> [1] Wen et al. "Principal Components Enable A New Language of Images." ICCV 2025.
>
> [2] Wu et al. "Towards Sequence Modeling Alignment between Tokenizer and Autoregressive Model."

---

> > ### Author Rebuttal · Reviewer_zBUU · 2026-04-03
> >
> > Thank you for the authors’ response and the additional experiments. I appreciate the time and effort invested in addressing the reviews. However, my main concerns still do not appear to be fully resolved.
> >
> > 1. 1D causal ViT decoder: I do not see sufficient empirical or theoretical analysis demonstrating that the causal design is particularly beneficial in this setting.
> > 2. Figure 3(a), Vanilla E2E variant: I am still unclear whether this result rules out the possibility that the model converges to a trivial solution.
> > 3. Reconstruction evaluation: Reconstruction FID alone does not seem sufficient to assess reconstruction quality. It would be more convincing to also include metrics such as SSIM and PSNR.
> > 4. ImageNet 512 experiments: As I understand it, results on ImageNet 512 are not provided.
> >
> > Again, I appreciate the authors’ efforts in preparing the response. At this stage, I am inclined to keep my current score, while also taking into account the perspectives of the other reviewers. Thank you.

---

> > > ### Author Response · Authors · 2026-04-08
> > >
> > > We thank the reviewer for the constructive feedback and prompt response.
> > >
> > > 1. **1D causal ViT decoder.** We would like to clarify that we do not claim the causal decoder as a main contribution of our paper, but rather a natural design choice following previous works [1,2]. Our main contribution is to enable the end-to-end training of 1D tokenizer and autoregressive models, which learns an autoregressive friendly latent space. In fact, our framework is agnostic to the specific design of model architecture, and our ablation study in the table above shows that using causal and bidirectional attention mask result in comparable performance, with trade-offs between reconstruction and generation quality.
> > >
> > > 2. **About the Vanilla E2E variant in Figure 3a**. Figure 3a shows that the vanilla E2E model is inclined to converge to a trivial solution, meaning using as few codes in the codebook as possible, because it is the easiest way to decrease the NTP loss. This is validated by both Figure 3a and Table 1, where the code usage of vanilla E2E, the rFID and gFID are much worse than the baseline. However, APR loss adds further constraints on the next-token prediction, and requires the reconstructed AR prediction to be close to the ground truth, and therefore rules out the possibility of the loss hacking issue. As shown in Figure 3 and Table 1, this prevents the model from hacking the NTP loss by lowering codebook usage, and achieves a much better  generation quality.
> > >
> > > 3. **Reconstruction evaluation.** Following your suggestion, we include more comprehensive reconstruction metrics as below. We include the results of recent discrete tokenizers for reference. Our tokenizer has a comparable performance on PSNR, SSIM and LPIPS despite using compact 1D compression, and achieves better a distributional performance measured by rFID.
> > >
> > >    | Models                      | PSNR  | SSIM | LPIPS | rFID |
> > >    | --------------------------- | ----- | ---- | ----- | ---- |
> > >    | EOSTok-L (n=4096)           | 22.15 | 0.67 | 0.231 | 0.73 |
> > >    | IBQ (n=16384)               | 22.01 | 0.61 | 0.224 | 1.37 |
> > >    | IBQ (n=262144)              | 22.69 | 0.64 | 0.203 | 1.00 |
> > >    | GigaTok-B-L (n=16384)       | 21.21 | 0.68 | 0.206 | 0.81 |
> > >    | LlamaGen (n=16384)          | 20.79 | 0.67 | 0.228 | 2.19 |
> > >    | Open-MAGVIT2 (n=16384)      | 20.79 | 0.62 | -     | 2.19 |
> > >    | Semanticist (continuous 1D) | 21.61 | 0.63 | -     | 0.78 |
> > >
> > > 4. **Larger-scale experiments.** We thank the reviewer for the suggestion. It is impossible to train the model to convergence in a short period, as it takes about 10 days on 8 H100 GPUs to train, but we are actively training our model now. While our training recipe takes 400 epochs in total, we report the intermediate results at 20 and 40 epochs here due to the time limit of the rebuttal phase. We compare it with TiTok, the only 1D tokenizer paper in Table 3 with ImageNet 512 results, as well as diffusion and mask AR baselines. As shown in the table below, EOSTok-L achieves 2.65 gFID without guidance at epoch 40, outperforming all baselines with many more training epochs. We will include full experimental results in the revised version.
> > >
> > > | ImageNet 512 | Epoch | gFID (w/o guidance) |
> > > | ------------ | ----- | ------------------- |
> > > | EOSTok-L     | 20    | 3.99                |
> > > | **EOSTok-L** | **40**    | **2.65**            |
> > > | TiTok-L-64   | 200   | 3.91                |
> > > | TiTok-B-128  | 200   | 4.17                |
> > > | DiT-XL/2     | 600   | 12.03               |
> > > | MaskDiT      | 800   | 10.79               |

---

### Official Review · Reviewer_f2LL · 2026-03-12

**Soundness:** 3
**Presentation:** 3
**Significance:** 3
**Originality:** 2
**Overall Recommendation:** 4
**Confidence:** 3

**Summary:**

This paper addresses several core challenges in image generation. Most existing models rely on tokenizers with a 2D grid structure, which introduces inherent bidirectional spatial dependencies that fundamentally mismatch the unidirectional decomposition (e.g., raster-scan order) required by autoregressive (AR) models. To resolve this, the paper proposes a 1D visual tokenizer that decouples 2D spatial priors. While previous methods typically employ a two-stage training process—training the tokenizer for image reconstruction first, then freezing it to train the AR model—the paper argues this hinders the tokenizer from learning representations better suited for generative tasks. Consequently, it introduces an End-to-End Joint Training paradigm.

**Compliance With Llm Reviewing Policy:**

Affirmed.

**Final Justification:**

After reviewing the author's responses and discussions, I hold a moderately **positive** stance toward this paper. The paper primarily proposes an end-to-end training method for tokenizers that aligns with semantic information, achieving promising results. The reason I did not give a strong accept is that both semantic alignment and end-to-end training have been previously proposed in works such as REPA and REPA-E.

**Key Questions For Authors:**

1. Compared to traditional two-stage training, what is the increase in Peak VRAM and total training duration (GPU Hours / FLOPs) for End-to-End Joint Training (e.g., for EOSTok-H with 1B parameters)? Figure 5b only compares loss curves but lacks a fair comparison under an equivalent compute budget.

2. What would the results look like if CFG or CFG interval strategies were applied?

3. Table 5 reveals a reconstruction-generation trade-off: increasing 1D sequence length improves reconstruction rFID but degrades AR generation quality (gFID). Current experiments are limited to ImageNet $256 \times 256$ (sequence length 256). If extended to megapixel resolutions, maintaining the same compression ratio would cause 1D sequence lengths to exceed 1024, making AR modeling extremely difficult. How do the authors evaluate the scalability of this 1D architecture and end-to-end framework for higher-resolution generation?

4. Section 3.3 uses DINOv2 for implicit and decoder alignment, proving it critical for AR accuracy and gFID reduction. To what extent does this success depend on the specific feature space of DINOv2 (which contains rich local/patch-level semantics)? How would performance change if a CLIP model, which focuses on global text alignment, were used instead? Is the framework robust to the choice of VFM?

**Limitations:**

The paper explicitly notes a trade-off between sequence length and AR generation quality. If the resolution scales to $512 \times 512$ or higher, the 1D sequence length will increase sharply, and the difficulty of AR modeling will rise exponentially. Since all experiments are currently limited to $256 \times 256$, the evidence for its effectiveness in ultra-high-resolution generation scenarios remains insufficient.

**Strengths And Weaknesses:**

Experimental Results
The model achieves a SOTA gFID of 1.48 without guidance. With guidance, gFID further drops to 1.38, with an Inception Score (IS) of 265.7. Even a smaller model, EOSTok-L (312M AR parameters), achieves a gFID of 1.74 without guidance, significantly outperforming mainstream 2D/1D AR models (e.g., LlamaGen, VAR) and masked models (e.g., TiTok, MaskGIT). That is good.

Originality
The methodology consists of "leveraging existing work" and "novel contributions":

Leveraged Components: The 1D tokenizer architecture follows a similar structure as TiTok (Yu et al., 2025b). The quantization module utilizes the existing IBQ. The concept of using VFM features like DINOv2 to assist generation is inspired by recent diffusion model works (e.g., REPA, VA-VAE). Strategies for regularizing token sequences are introduced from prior works like FlexTok and Semanticist.

Novel Contributions: This is the first work to jointly optimize a 1D visual tokenizer and an AR generative model from scratch in a single stage, allowing gradients from discrete token predictions to backpropagate to the tokenizer. Since using only NTP loss during joint training causes "latent space collapse," the authors propose a new loss that decodes predicted tokens back into pixel space to compute errors, bridging the gap between discrete prediction and pixel quality. Additionally, global semantics from VFMs (e.g., DINOv2) are injected into the 1D tokenizer. To avoid breaking 1D AR properties via direct alignment, the authors innovatively align hidden patch embeddings from the encoder's intermediate layers rather than the final 1D tokens.

---

> ### Author Rebuttal · Authors · 2026-03-31
>
> We thank the reviewer for the detailed and constructive feedback, and for recognizing the strong experimental results and novel contributions of our work. We address each question below.
>
> 1. **Computation and memory cost.** Our EOSTok training scheme combines conventional two-stage training into one end-to-end training stage (Figure 1). Since we train both the tokenizer and AR generator in the same stage, each training step of EOSTok indeed requires more computation than the second stage of conventional AR training. However, the total computation budget of EOSTok is about the same as two-stage tokenizer and AR generator training. For Figure 5b, the tokenizer for two-stage training is already pretrained for 1 million steps, making the total computation budget comparable. The only additional overhead in terms of total computation is the APR loss, which costs one extra VAE decoding per step; this decoding is also parallelized with the original VAE decoding for reconstruction loss.
>
>    We further provide detailed cost analysis of EOSTok, including GFLOPs of each module of EOSTok, and the peak memory usage in training. We use 8 A100 GPUs for training, which takes up to 10 days for training the largest EOSTok model while using a batch size of 32 per device (256 globally). We will make this clear in our revision.
>
>    |                                                     | EOSTok-L              | EOSTok-H                |
>    | --------------------------------------------------- | --------------------- | ----------------------- |
>    | Tokenizer GFLOPs                                    | 91 Encode + 91 Decode | 210 Encode + 210 Decode |
>    | AR generator GFLOPs                                 | 162                   | 338                     |
>    | VFM embedder GFLOPs                                 | 162                   | 162                     |
>    | **Forward total (AR + Encode + 2x Decode + VFM)**   | **597**               | **1130**                |
>    | **Overhead compared to two-stage training**         | **15.2%**             | **18.6%**               |
>    | **Training Peak memory (batch size per device=32)** | **36.51 GB**          | **56.96 GB**            |
>
>
>
> 2. **CFG strategy.** We apply a classifier-free guidance strategy to our EOSTok models on the logits of the AR generator, i.e., $\ell = \ell_u + s(\ell_c - \ell_u)$. We adopt the pow-cosine guidance schedule for generating sequences of length 256, following the practice of [1], with a CFG scale of 1.8 and a power of 1. The generation results with guidance applied are listed in Table 3.
>
> 3. **Trade-off between reconstruction and generation, and high-resolution generation.** We report the trade-off between reconstruction and generation in Table 6. This phenomenon is not unique to 1D tokenization. It has also been observed in many existing works, from [2] on 2D latent diffusion models to [3] with 1D tokenizers, where decreasing the latent dimension or length improves generation quality. Therefore, the canonical way to scale EOSTok to higher resolutions is to keep the same latent sequence length of 256 for generation.
>
>    Moreover, as shown in Table 5, using as few as 64 tokens is able to achieve good generation quality on ImageNet 256, which suggests that 256 tokens for ImageNet 512 would be sufficient for good generation quality as well. This is also corroborated by [3], where 128 tokens suffice for ImageNet 512 generation. In our paper, we benchmark EOSTok models on ImageNet 256 because most baseline methods in Table 3 only report generation on ImageNet 256, but in principle EOSTok can be directly applied to ImageNet 512 training as well.
>
> 4. **Robustness to choice of VFM.** We further experiment with EOSTok-L in our ablation setting using a SigLIP2 [4] model, which contains richer global semantic information. As shown in the table below, SigLIP2 slightly improves generative results compared to DINOv2, which shows the robustness of our framework to the choice of pretrained vision foundation models.
>
> |                     | EOSTok-L with DINOv2 | EOSTok-L with SigLIP2 |
> | ------------------- | -------------------- | --------------------- |
> | gFID (w/o guidance) | 3.32                 | 3.02                  |
> | rFID                | 1.02                 | 0.88                  |
>
> [1] Yu et al. "Randomized autoregressive visual generation." ICCV 2025.
>
> [2] Yao et al. "Reconstruction vs. Generation: Taming Optimization Dilemma in Latent Diffusion Models." CVPR 2025.
>
> [3] Yu et al. "An Image is Worth 32 Tokens for Reconstruction and Generation." arXiv: 2406.07550.
>
> [4] Tschannen et al. "SigLIP 2: Multilingual Vision-Language Encoders with Improved Semantic Understanding, Localization, and Dense Features." arXiv: 2502.14786.

---

> > ### Author Rebuttal · Reviewer_f2LL · 2026-04-03
> >
> > The author's response has further increased my concerns and confusion.
> >
> > Regarding line 318-319 and Appendix A.3, it is stated that autoguidance was employed for EOSTok-L and EOSTok-H. However, my original inquiry pertained to the results when using Classifier-Free Guidance (CFG) or CFG intervals. I am interested in understanding the performance of these two models specifically under CFG settings.
> >
> > The author's statement that EOSTok can, in principle, be directly applied to ImageNet 512 training does not address my concerns; it remains a theoretical assertion rather than empirical evidence.
> >
> > I would appreciate it if the authors could provide the following:
> >
> > What would be the performance of EOSTok-L and EOSTok-H in Table 3 if CFG or CFG intervals were applied?
> >
> > Preliminary experimental results on ImageNet 512 or text-to-image tasks. Without such evidence, the generalizability of the proposed method remains questionable.

---

> > > ### Author Response · Authors · 2026-04-08
> > >
> > > We thank the reviewer for the constructive feedback and prompt response. We would like to provide further clarification regarding the concerns.
> > >
> > > 1. **CFG results.** Sorry for causing the confusion. We specifically apply Autoguidance for EOSTok-L and EOSTok-H as we find a diminishing effect on applying vanilla classifier-free guidance as the unconditional generation improves. We find that applying a CFG scale as low as 1.2 or 1.1 yields better results, whereas further increasing the CFG scale harms sampling diversity.
> > >
> > >    | gFID                        | EOSTok-L | EOSTok-H |
> > >    | --------------------------- | -------- | -------- |
> > >    | w/o guidance                | 1.74     | 1.48     |
> > >    | Vanilla classifier guidance | 1.46     | 1.45     |
> > >    | Autoguidance                | 1.35     | 1.38     |
> > >
> > > 2. **Larger-scale experiments.** We thank the reviewer for the suggestion. It is impossible to train the model to convergence in a short period, as it takes about 10 days on 8 H100 GPUs to train, but we are actively training our model now. While our training recipe takes 400 epochs in total, we report the intermediate results at 20 and 40 epochs here due to the time limit of the rebuttal phase. We compare it with TiTok, the only 1D tokenizer paper in Table 3 with ImageNet 512 results, as well as diffusion and mask AR baselines. As shown in the table below, EOSTok-L achieves 2.65 gFID without guidance at epoch 40, outperforming all baselines with many more training epochs. We will include full experimental results in the revised version.
> > >
> > > | ImageNet 512 | Epoch | gFID (w/o guidance) |
> > > | ------------ | ----- | ------------------- |
> > > | EOSTok-L     | 20    | 3.99                |
> > > | **EOSTok-L** | **40**    | **2.65**            |
> > > | TiTok-L-64   | 200   | 3.91                |
> > > | TiTok-B-128  | 200   | 4.17                |
> > > | DiT-XL/2     | 600   | 12.03               |
> > > | MaskDiT      | 800   | 10.79               |

---

### Decision · Program_Chairs · 2026-04-30

**Decision:**

Accept (spotlight)

**Comment:**

This paper proposes EOSTok, an end-to-end framework that jointly trains a 1D discrete tokenizer with an autoregressive generator via an APR loss and VFM semantic alignment. Scores are 4/5/4/5 (mean 4.5). Reviewers broadly agree on the practical significance and strong empirical results on ImageNet 256×256. The main reservations—raised by f2LL and m9oV and not fully resolved after discussion—are that the novelty over REPA and REPA-E is incremental, and that the theoretical motivation for the APR loss, training stability analysis, and large-scale (512) validation remain underdeveloped. These are acknowledged limitations but do not undermine the core contribution. I recommend acceptance, with the expectation that the final version better differentiates from REPA/REPA-E, strengthens the APR loss analysis, and incorporates the full ImageNet-512 results committed to during rebuttal.